# Temporal transitions of spontaneous brain activity

Zhiwei Ma[†], Nanyin Zhang*

Department of Biomedical Engineering, The Huck Institutes of Life Sciences, The Pennsylvania State University, State College, United States

**Abstract** Spontaneous brain activity, typically investigated using resting-state fMRI (rsfMRI), provides a measure of inter-areal resting-state functional connectivity (RSFC). Although it has been established that RSFC is non-stationary, previous dynamic rsfMRI studies mainly focused on revealing the spatial characteristics of dynamic RSFC patterns, but the temporal relationship between these RSFC patterns remains elusive. Here we investigated the temporal organization of characteristic RSFC patterns in awake rats and humans. We found that transitions between RSFC patterns were not random but followed specific sequential orders. The organization of RSFC pattern transitions was further analyzed using graph theory, and pivotal RSFC patterns in transitions were identified. This study has demonstrated that spontaneous brain activity is not only nonrandom spatially, but also nonrandom temporally, and this feature is well conserved between rodents and humans. These results offer new insights into understanding the spatiotemporal dynamics of spontaneous activity in the mammalian brain.
DOI: https://doi.org/10.7554/eLife.33562.001

*For correspondence:
nuz2@psu.edu

Present address: [†]Laboratory of Functional and Molecular Imaging, National Institute of Neurological Disorders and Stroke, Bethesda, United States

**Competing interests:** The authors declare that no competing interests exist.

## Introduction

Multiple lines of evidence indicate that spontaneous brain activity plays an essential role in brain function (*Raichle and Mintun, 2006*; *Zhang and Raichle, 2010*). For instance, intrinsic neuronal signaling consumes the vast majority of brain energy (*Raichle, 2006*, *2010*). Investigation of spontaneous brain activity, predominantly conducted using resting-state functional magnetic resonance imaging (rsfMRI) (*Biswal et al., 1995*; *Fox and Raichle, 2007*), has provided critical insight into the intrinsic organization of the brain network. Using spontaneously fluctuating blood-oxygenation-level dependent (BOLD) signal measured by rsfMRI, resting-state functional connectivity (RSFC) between brain regions can be gauged by statistical interdependence of their rsfMRI signals over the period of data acquisition (*Fox and Raichle, 2007*). Based on this quantity, multiple brain networks of functionally-related regions have been identified in both humans and animals, which convey the information of stable functional connectivity organization of the brain (*Beckmann et al., 2005*; *Fox et al., 2005*; *Damoiseaux et al., 2006*; *Smith et al., 2009*; *Allen et al., 2011*; *Liang et al., 2011*).

Conventional rsfMRI studies generally focus on steady features of RSFC by assuming that RSFC is stationary during the acquisition period. However, meaningful temporal variability of RSFC at shorter time scales has also been discovered (*Chang and Glover, 2010*). This initial research and its follow-up studies revealed dynamic properties of RSFC, indicating that the stationarity assumption of RSFC would be overly simplistic for understanding spontaneous brain activity (*Hutchison et al., 2013a*; *Preti et al., 2017*; *Chang et al., 2016a*). Indeed, using sliding window analysis and clustering methods, temporally alternating but spatially repeatable RSFC patterns have been identified (*Allen et al., 2014*). In addition, *Liu and Duyn (2013)* developed a method that examined instantaneous co-activations of BOLD signal at single rsfMRI frames and found that BOLD co-activation patterns well corresponded brain connectivity configurations (*Liu and Duyn, 2013*). With this method, the default mode network, which is a single network under the assumption of stationary RSFC, can be

decomposed into multiple sub-networks with distinct spatiotemporal characteristics and functional relevance (*Liu and Duyn, 2013*). Notably, the neurophysiologic relevance of dynamic RSFC has been validated in multiple studies using simultaneous electrophysiology and rsfMRI acquisitions (*Tagliazucchi et al., 2012*; *Chang et al., 2013*; *Keilholz, 2014*; *Liu et al., 2015b*).

In parallel with blossoming dynamic RSFC studies in humans, dynamic RSFC studies in animal models have also been conducted. Animals' brain preserves fundamental organizational properties as the human brain (*Liang et al., 2011*; *Ma et al., 2016*), and can serve as a translational model for studying complicated brain dynamics. Using either the sliding window or co-activation pattern approach, dynamic RSFC patterns have been found in both awake and anesthetized rats, as well as in anesthetized monkeys (*Majeed et al., 2011*; *Hutchison et al., 2013b*; *Keilholz et al., 2013*; *Mohajerani et al., 2013*; *Liang et al., 2015*; *Grandjean et al., 2017*; *Ma et al., 2017*). These results suggest that dynamics in RSFC might be a general feature in the mammalian brain.

Despite the critical advancement, aforementioned dynamic rsfMRI studies have mainly focused on revealing the *spatial* characteristics of RSFC patterns that were non-stationary, while the *temporal relationship* between these RSFC patterns is still unclear. Particularly, although the existence of temporal transitions between characteristic RSFC patterns has been established, it remains elusive whether these transitions are random or organized in an orderly manner (*Majeed et al., 2011*; *Zalesky et al., 2014*; *Mitra et al., 2015*; *Preti et al., 2017*). Lack of such information highlights a gap in elucidating the temporal relationship of separate brain connectivity configurations, and thus hinders the comprehensive characterization of spatiotemporal dynamics of spontaneous brain activity.

To address this issue, in the present study we studied the temporal transitions of intrinsic brain activity in both awake rats and humans. The reproducibility of the RSFC pattern transitions was examined. In addition, the organization of RSFC pattern transitions in rats and humans were respectively studied using graph theory analysis. RSFC patterns that were pivotal in temporal transitions were further identified.

## Results

In this study, we investigated the temporal transitions between spontaneous brain activity patterns in awake rats and humans. In rat data, we first obtained a library of 40 characteristic RSFC patterns, using seed-based correlational analysis with seeds defined by parcels in a whole-brain RSFC-based parcellation (*Ma et al., 2016*). These characteristic RSFC patterns were used as the reference patterns. Subsequently, based on the notion that the BOLD co-activation patterns of single rsfMRI frames represent their RSFC patterns (*Liu et al., 2013*; *Liu and Duyn, 2013*), each rsfMRI frame was matched to one of the 40 reference RSFC patterns that had the highest spatial similarity to the BOLD co-activation pattern of the frame. This step generated a time sequence of characteristic RSFC patterns for each rsfMRI run. Temporal transitions between every pair of RSFC patterns were then counted, which created a RSFC pattern transition matrix. A weighted directed transition network was constructed by thresholding this transition matrix, and was analyzed using graph theory. The same approach was also applied to human rsfMRI data to examine the translational value of the findings in rats. A schematic illustration of these procedures is shown in *Figure 1*.

### Characteristic RSFC patterns in the awake rat brain

An example of a characteristic RSFC pattern is shown in *Figure 2*, and the other 39 characteristic RSFC patterns are shown in *Figure 2—figure supplements 1–5*. As the whole-brain parcellation scheme we adopted maximized within-parcel and minimized cross-parcel RSFC profile similarity, these 40 group-level seed-based RSFC maps represented a set of characteristic RSFC patterns in the awake rat brain, and were used as the reference patterns. Notably, the number 40 was arbitrarily selected as an example of low-dimensionality parcellation of the rat brain. Similar analysis can be applied using other parcel numbers.

*Figure 2* (right panel) also shows the averaged pattern of rsfMRI frames that were matched to the reference RSFC pattern, which demonstrated high reminiscence between the BOLD co-activation pattern of single rsfMRI frames and the RSFC pattern it corresponded to (correlation coefficient = 0.91).

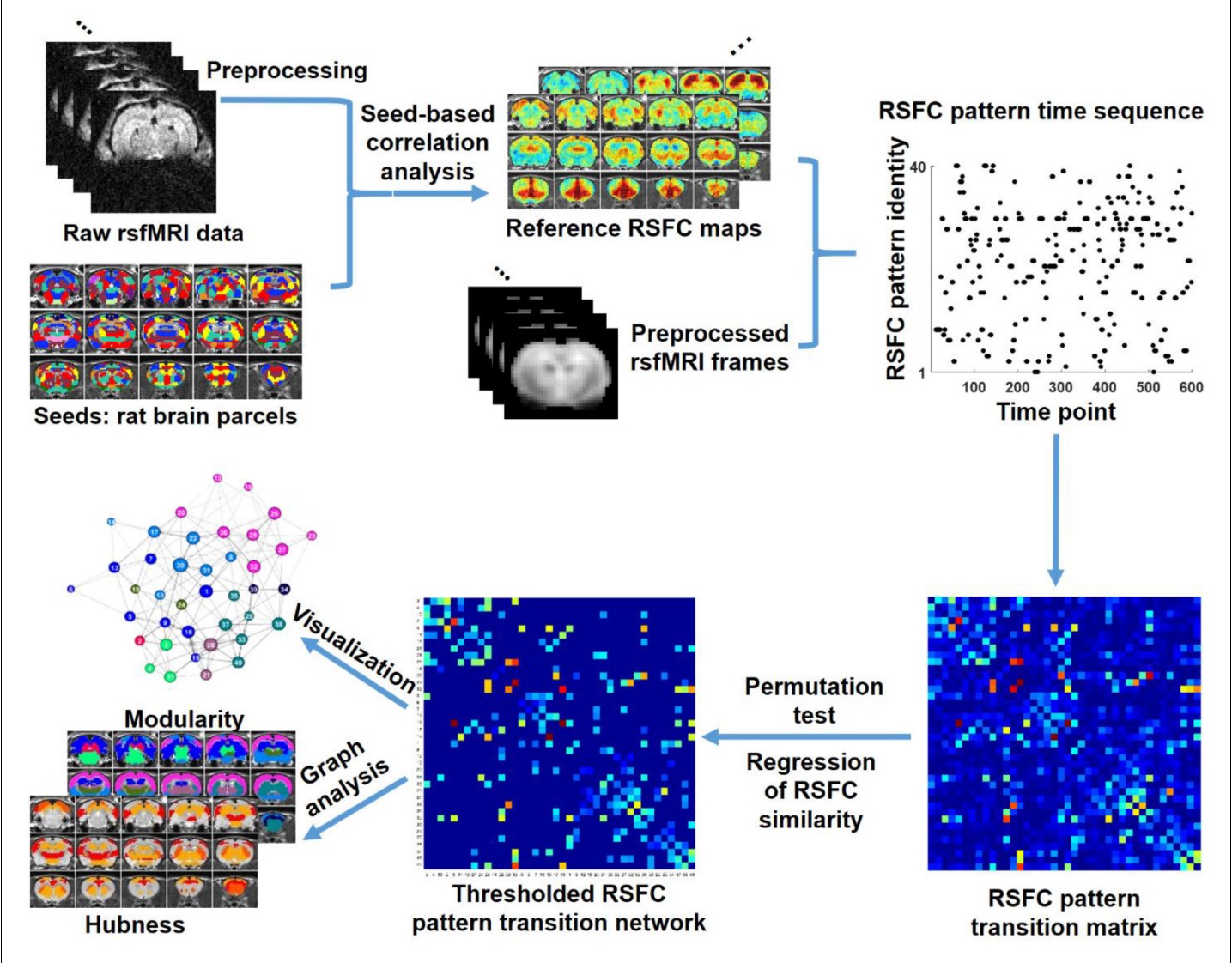

**Figure 1.** Schematic illustration of the data analysis pipeline.
DOI: https://doi.org/10.7554/eLife.33562.002

## Reproducible temporal transitions between RSFC patterns

We first demonstrated that temporal transitions between RSFC patterns were highly reproducible at the group level. We randomly split all rats into two subgroups and obtained the transition matrix for each subgroup. Both matrices exhibited high similarity (*Figure 3a*), reflected by a significant correlation (r = 0.86, p ≈ 0) between the corresponding off-diagonal entries. To control for the possible bias that transitions between similar RSFC patterns may have a higher chance to occur in both subgroups, which can inflate the reproducibility, we regressed out the spatial similarities between reference RSFC patterns from both transition matrices. The reproducibility remained high after regression, with a significant correlation value of 0.77 (p ≈ 0, *Figure 3b*). Taken together, these results suggest that transitions between RSFC patterns are not random but follow specific temporal sequences in awake rats, and these transition sequences are not dictated by the similarity between RSFC patterns.

To further examine whether reproducible RSFC pattern transitions were dominated by a small portion of rats, we assessed the reproducibility of RSFC pattern transitions for each individual animal by computing Pearson correlation between each individual-level transition matrix and the group-

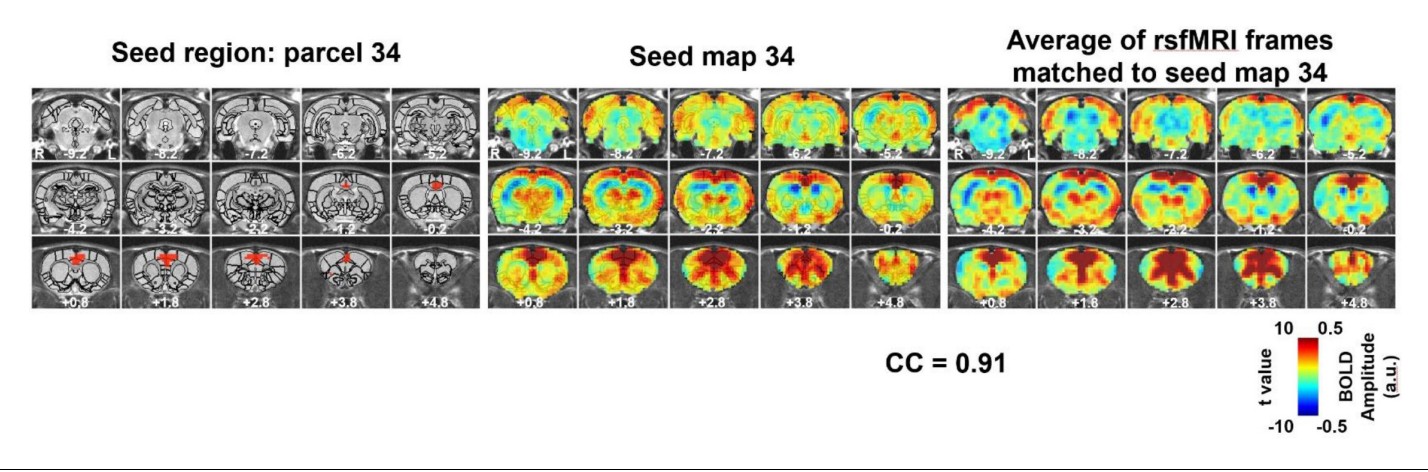

**Figure 2.** An example of a RSFC spatial pattern.  Left: seed region; Mid: RSFC pattern of the seed. Color bar indicates t values; Right: average of rsfMRI frames matched to the RSFC pattern. Color bar shows BOLD amplitude. Distance to bregma is listed at the bottom of each slice. CC: spatial correlation coefficient between the average of rsfMRI frames (Right) and the corresponding seed map (Mid).

DOI: https://doi.org/10.7554/eLife.33562.003

The following figure supplements are available for figure 2:

**Figure supplement 1.** Characteristic RSFC patterns (1-8) in the awake rat brain.
DOI: https://doi.org/10.7554/eLife.33562.004
**Figure supplement 2.** Characteristic RSFC patterns (9-16) in the awake rat brain.
DOI: https://doi.org/10.7554/eLife.33562.005
**Figure supplement 3.** Characteristic RSFC patterns (17-24) in the awake rat brain.
DOI: https://doi.org/10.7554/eLife.33562.006
**Figure supplement 4.** Characteristic RSFC patterns (25-32) in the awake rat brain.
DOI: https://doi.org/10.7554/eLife.33562.007
**Figure supplement 5.** Characteristic RSFC patterns (33, 35-40) in the awake rat brain.
DOI: https://doi.org/10.7554/eLife.33562.008

level transition matrix. Fisher Z-transformed correlation values were then averaged across rats. Our data showed a significant individual-level reproducibility (mean (±SD)=0.57 (±0.14), p ≈ 0). These results collectively indicate that nonrandom RSFC pattern transitions are a characteristic feature in awake rats.

To rule out the possibility that RSFC pattern transitions were caused by head motion (*Laumann et al., 2016*), we conducted several additional analyses. First, we re-evaluated the reproducibility of RSFC transitions between two subgroups of rats with relatively high and low motion, respectively. Rats in the first subgroup all had the motion level below the median, quantified by framewise displacement (FD). Rats in the second subgroup all had the motion level above the median. The mean (±SD) FD of the below- and above-median subgroups were 0.037 (±0.024) mm and 0.054 (±0.033) mm, respectively. Transition matrices were obtained in these two subgroups, respectively. Comparing these two transition matrices yielded a reproducibility of 0.786 with regression of RSFC pattern similarities, and 0.872 without regression of RSFC pattern similarities (*Figure 3—figure supplement 1*), which is similar to the reproducibility assessed based on a random division to subgroups (0.77 and 0.86 with and without regression of RSFC pattern similarities, respectively, *Figure 3*). In addition, the transition matrices from both subgroups were highly consistent with those in subgroups randomly divided (*Figure 3*). To statistically test whether the reproducibility using motion-based division to subgroups was different from that based on random divisions, we repeated random subgroup divisions 10000 times. Mean reproducibility (±SD) across all 10000 trials was 0.876 (±0.010) and 0.791 (±0.017) without and with regression of RSFC pattern similarities, respectively. *Figure 3—figure supplement 2* shows the distributions of the reproducibility between randomly divided subgroups across all trials. This data demonstrated that the reproducibility using motion-based division to subgroups was not statistically different from the reproducibility based on

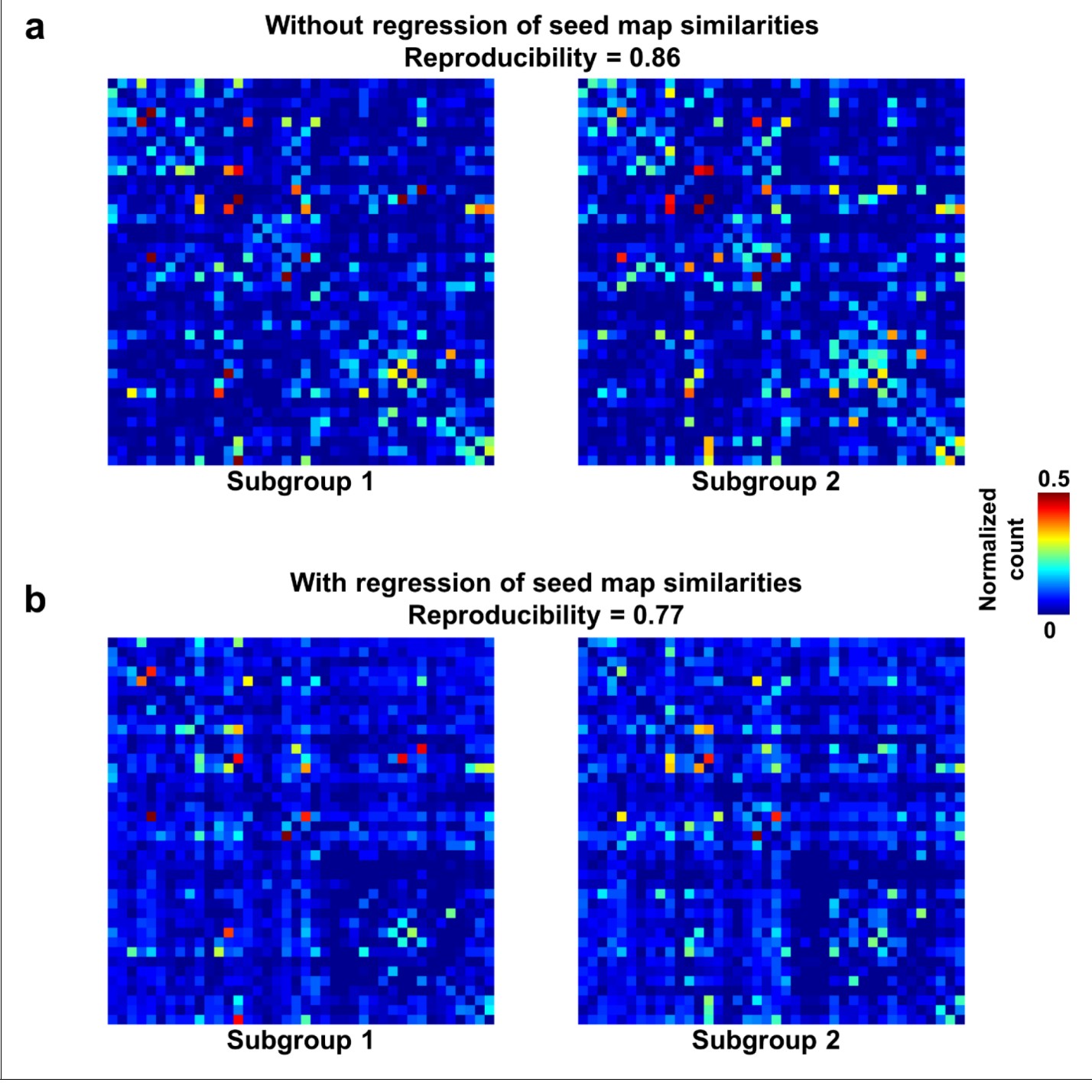

**Figure 3.** Reproducibility of the RSFC pattern transitions. (a) RSFC pattern transition matrices of subgroups 1 and 2 without regression of spatial similarities between reference RSFC patterns. (b) RSFC pattern transition matrices of subgroups 1 and 2 with regression of spatial similarities between reference RSFC patterns. Entries in each transition matrix were normalized to the range of [0, 1].

DOI: https://doi.org/10.7554/eLife.33562.009

The following figure supplements are available for figure 3:

**Figure supplement 1.** Reproducibility of RSFC pattern transitions between two subgroups of rats with relatively high (FD above median, right panels) and low (FD below median, left panels) motion levels without (top panels) and with (bottom panels) regression of RSFC pattern similarities.

DOI: https://doi.org/10.7554/eLife.33562.010

**Figure supplement 2.** Distribution of the reproducibility of RSFC pattern transition matrices between randomly divided subgroups across 10000 trials.

DOI: https://doi.org/10.7554/eLife.33562.011

*Figure 3 continued*

**Figure supplement 3.** RSFC pattern transitions at different motion censoring threshold.

DOI: https://doi.org/10.7554/eLife.33562.012

random division regardless whether RSFC pattern similarities were regressed out (p=0.75) or not (p=0.67). These results indicate that high reproducibility in RSFC pattern transitions was not attributed to head motion.

In the second analysis, we directly compared the motion level between rsfMRI frames involved in RSFC pattern transitions versus those that were not in transitions. All rsfMRI frames analyzed were categorized into two groups. The first group included frames whose preceding and/or successive frame corresponded to a different RSFC pattern (i.e. in transitions). The second group included frames whose preceding and successive frames were the same RSFC pattern (i.e. not in transition). These two groups of rsfMRI frames showed consistent motion levels, quantified by their FD values (p=0.44, two-sample t-test), again indicating that RSFC pattern transitions were not triggered by head motion. To further test whether some RSFC pattern transitions were induced by head motion, we measured the head motion during each transition and compared the mean head motion level for each transition sequence and the occurrence count of this transition sequence. Specifically, we calculated the mean FD for transitions between every two RSFC patterns. This calculation yielded a $40 \times 40$ matrix, in which each element quantified the mean FD for each transition sequence (e.g. element (i,j) of this matrix measured the mean FD for the transition from RSFC pattern i to RSFC pattern j). Our data showed that the correlation between this transition FD matrix and the RSFC pattern transition matrix was minimal (r = −0.034), suggesting that the mean head motion level during each transition sequence did not predict the occurrence count of this transition sequence. This result further supported that RSFC pattern transitions were independent of head motion, and RSFC transitions did not follow head motion.

To examine whether our results were dependent on the motion censoring threshold selected (FD <0.2 mm), we reanalyzed our data using a more stringent censoring threshold (FC <0.1 mm). At this threshold and also keeping all other motion control criteria identical, very similar RSFC pattern transition matrices were obtained (*Figure 3—figure supplement 3*). The correlations between the RSFC pattern transition matrices at FD <0.1 mm and those at FD <0.2 mm were 0.83 and 0.88 with and without regression of RSFC pattern similarities, respectively, suggesting that our results were robust and insensitive to the motion censoring threshold applied.

## Within- and between-brain system transitions

*Figure 4* shows the group-level transition matrix thresholded using a permutation test (p<0.05, FDR corrected). Rows/columns in the transition matrix were arranged based on the brain system that the seed region of the reference RSFC pattern belonged to. Transitions between RSFC patterns tended to occur within the same brain system, as shown by a relatively denser distribution of near-diagonal nonzero elements in the matrix. However, cross-system transitions such as striatal-thalamic, striatal-somatosensory, striatal-prefrontal, striatal-hippocampal, hippocampal-amygdala, amygdala-motor transitions were observed.

## Organization of RSFC pattern transitions

A directed weighted graph of the RSFC pattern transition network was constructed based on the group-level thresholded transition matrix (*Figure 4*), as shown in *Figure 5*. The number of edges was 242, yielding a connection density of 15.5%. The transition network exhibited a prominent community structure with nine modules identified using the Louvain community detection algorithm (*Vincent et al., 2008*), suggesting that RSFC patterns belonging to the same modules had a higher probability to transit between each other than RSFC patterns across modules. The corresponding seed regions of RSFC patterns were color coded based on the community affiliations (*Figure 5* inlet). Module one primarily covered hippocampal and retrohippocampal networks as well as caudal visual networks. Module two included caudal midbrain networks. Module three was comprised of brainstem and rostral midbrain networks. Module four covered rostral visual, amygdala, hypothalamic as well as motor and olfactory networks. Module five was dominated by auditory and somatosensory

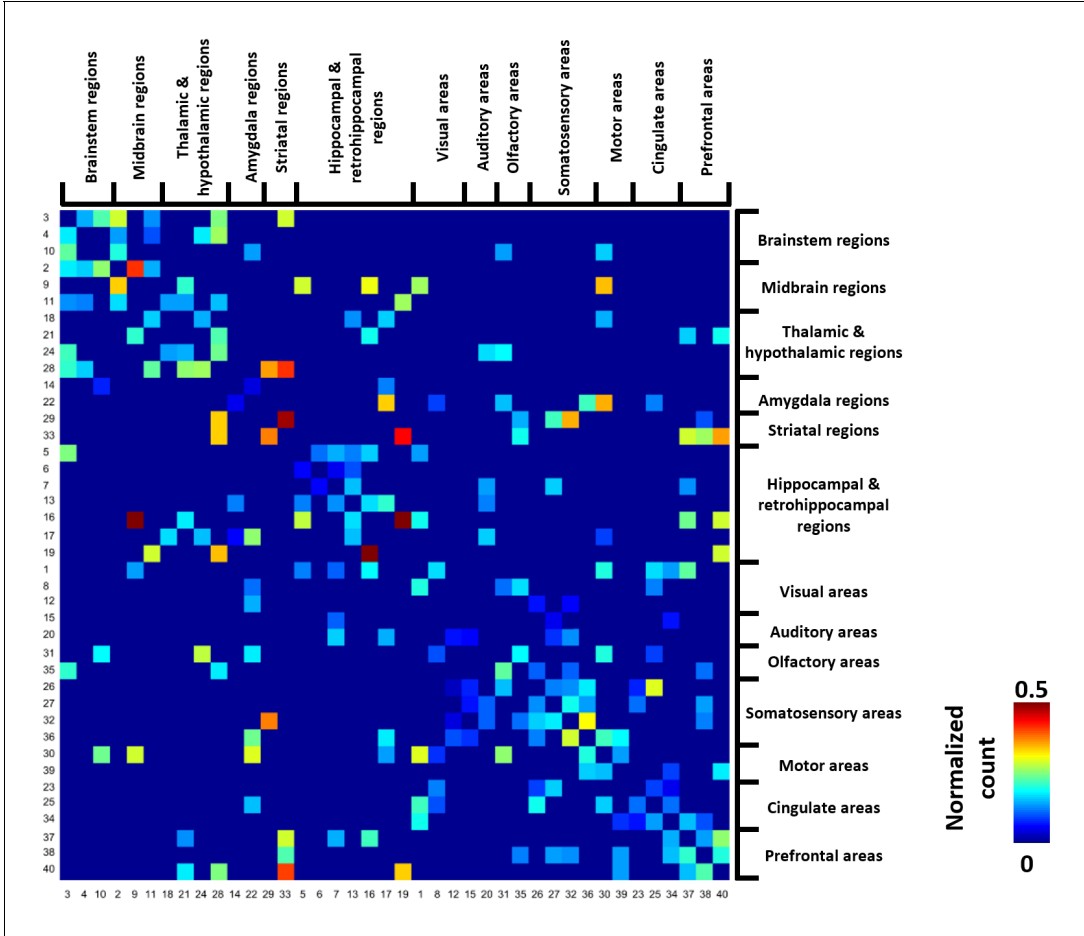

**Figure 4.** Thresholded group-level RSFC pattern transition matrix after regression of RSFC pattern similarities. Rows/columns are arranged based on the brain system of the seed regions. Numbers next to/below rows/columns correspond to the seed map number in *Figure 2* and *Figure 2—figure supplement 1*.

DOI: https://doi.org/10.7554/eLife.33562.013

networks. Module six captured posterior ventral thalamic networks. Module seven included anterior thalamic networks. Module eight covered striatal and prefrontal networks. Module nine mainly included anterior cingulate cortex network. Networks from the same system usually fell into the same community, again indicating transitions between RSFC patterns frequently occurred within the same brain system. However, networks from different systems were also observed in the same modules, which highlights the importance of cross-system transitions.

By quantifying the node-specific graph measures of node strength, betweenness centrality, characteristic path length and local clustering coefficient, hub nodes (i.e. pivotal RSFC patterns) in the graph were identified. Six RSFC patterns were identified as hubs in rats (hub score $\geq$3) including the networks of retrosplenial cortex, dorsal superior and inferior colliculi, hippocampus, anterior ventral thalamus, striatum and motor cortex. *Figure 6* shows the seed regions of RSFC patterns with hub score $\geq$1, color coded based on the hub score. No color was given to seed regions of RSFC patterns with hub score = 0.

*Figure 7* shows RSFC pattern transitions of four representative hubs (red nodes), demonstrating the pivotal role of these patterns in RSFC temporal transitions. The majority of transitions between hubs and other RSFC patterns were bidirectional. *Figure 7a* shows transitions of the hub network of the superior and inferior colliculi with the networks of the periaqueductal gray, hippocampus, dorsal thalamus, hypothalamus, caudal visual cortex and motor areas. *Figure 7b* demonstrates the transitions between the hippocampus network (hub) and the RSFC networks of the superior and inferior colliculi, hippocampus/retrohippocampus, caudal visual cortex, as well as prefrontal and orbital

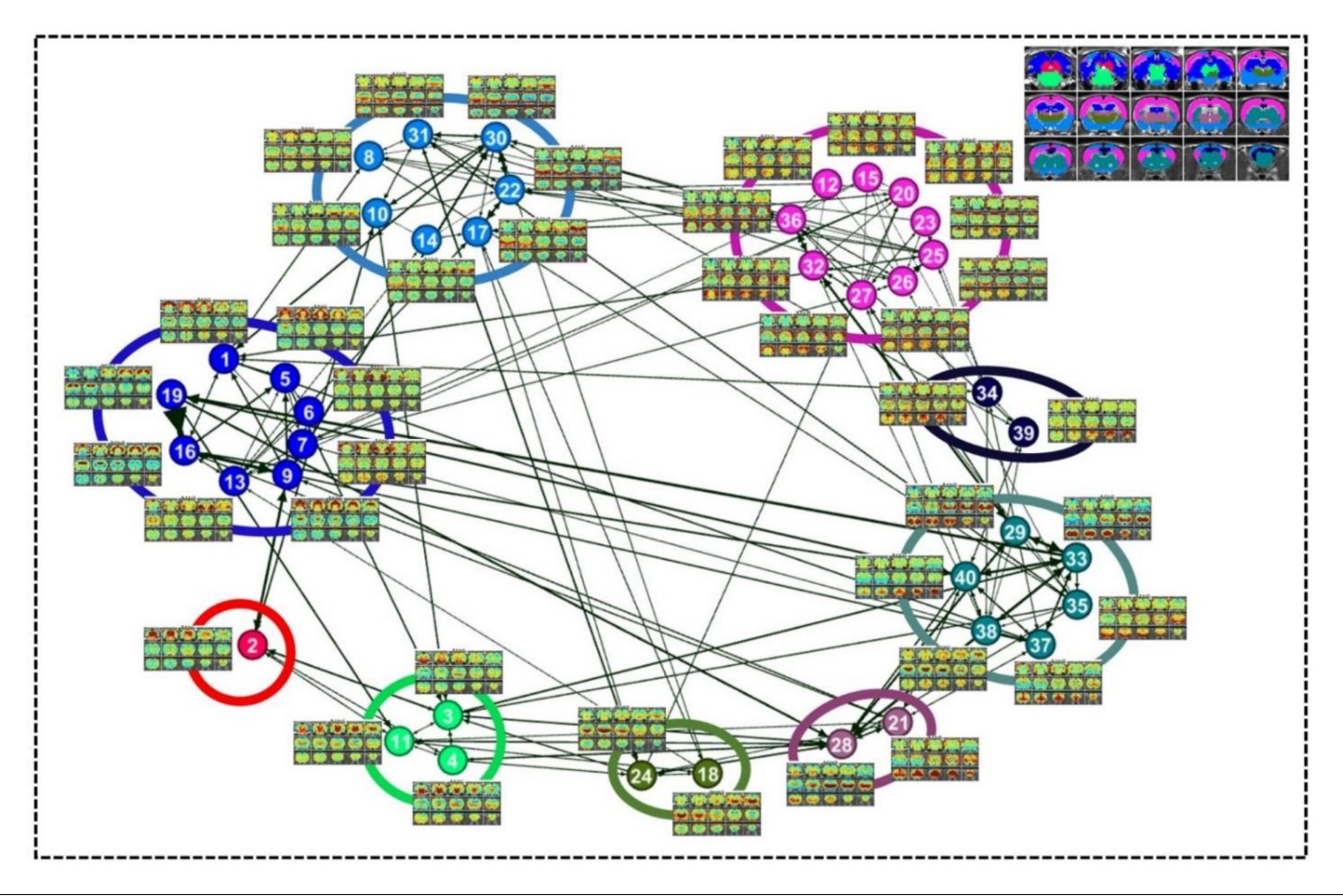

**Figure 5.** Community structures of the RSFC pattern transition network. The thresholded transition matrix in *Figure 4* was used as the adjacency matrix to generate a directed weighted graph. The layout of nodes was based on a force-field algorithm (*Jacomy et al., 2014*). The node number corresponds to the seed map number in *Figure 2* and *Figure 2—figure supplement 1—5*. Inlet: seed regions of RSFC patterns colored coded based on the community affiliations of nodes (i.e. RSFC patterns).
DOI: https://doi.org/10.7554/eLife.33562.014

cortices. *Figure 7c* displays transitions between the hub network of the anterior ventral thalamus and the RSFC networks of the brainstem, midbrain, dorsal CA1, dorsal thalamus, posterior ventral thalamus, dorsal caudate-putamen (CPu), olfactory tubercle, and orbital cortex. *Figure 7d* illustrates the transitions between the hub network of the ventral CPu, and the RSFC networks of the brainstem and olfactory tubercle, as well as infralimbic, prelimbic, and orbital cortices. Taken together, these results indicate that hub RSFC patterns were centralized patterns that play a pivotal role in transitions with other RSFC patterns involving multiple brain systems.

## RSFC pattern transitions in humans

To assess whether temporal transitions between RSFC patterns were also nonrandom in humans, we applied the same analysis to rsfMRI data from 812 human subjects in the HCP. Each frame was matched to one of 333 characteristic RSFC patterns defined by a well-established RSFC-based parcellation in humans (*Gordon et al., 2016*), and the number of transitions between every two RSFC patterns was counted for each subject. All subjects were then randomly split into two subgroups (406 subjects in each subgroup). The reproducibility between two subgroups was 0.9955 (without regression of seed map similarities, *Figure 8a*), and 0.9954 (with the regression of seed map similarities, *Figure 8b*). To assess the reproducibility at the individual level, the correlation between the transition matrix of each individual subject versus the group-level transition matrix was calculated.

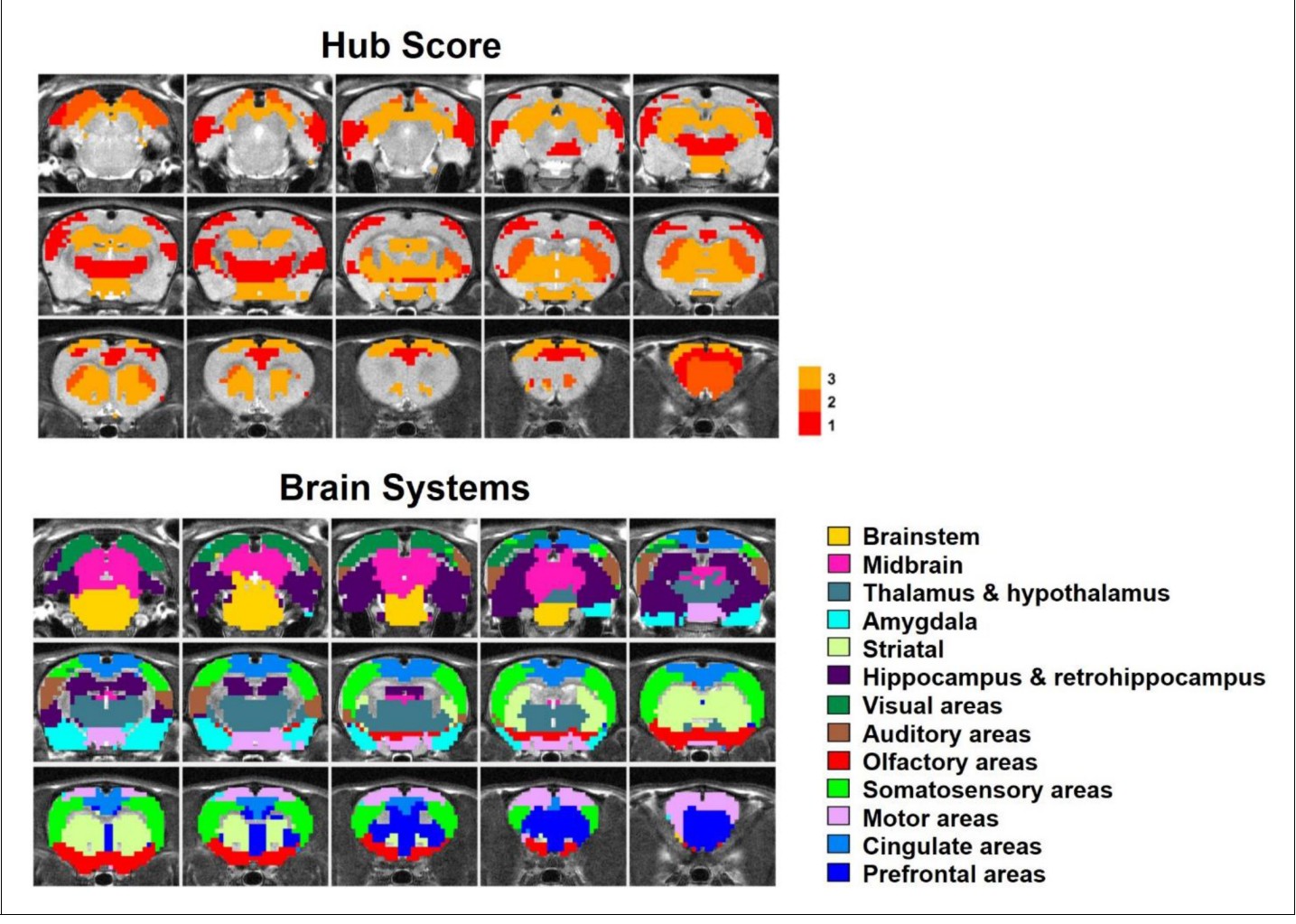

**Figure 6.** Top: Seed regions of RSFC patterns with hub score ≥1, color coded based on the hub score. No color was given to seed regions of RSFC patterns with hub score = 0. Bottom: the brain systems of seed regions.
DOI: https://doi.org/10.7554/eLife.33562.015

The mean correlation (±SD) across all subjects was 0.60 (±0.05). All these results were highly consistent with our findings in awake rats, suggesting that nonrandom transitions between RSFC patterns are conserved across species and might represent a characteristic feature of the mammalian brain.

The group-level transition matrix obtained from all 812 human subjects was thresholded using the same permutation test ($p < 0.05$, FDR corrected), as shown in *Figure 9*. Rows/columns of the RSFC transition matrix were grouped based on the brain system of the seed region. Similar to RSFC pattern transitions in rats, transitions between RSFC patterns in humans tended to occur within the same brain system, indicated by a denser distribution of near-diagonal nonzero elements in the matrix. However, considerable across-system RSFC pattern transitions were also evident.

The hub score of each RSFC pattern was calculated in the same way as the rat data. *Figure 10* showed seed regions of RSFC patterns with hub score ≥1, color coded based on the hub score. Seed regions of RSFC patterns with hub score = 0 were not given any color. Our results demonstrated that the human RSFC pattern transitions contain multiple hubs (hub score ≥3) in separate brain systems including default-mode (five nodes), cingulo-opercular (eight nodes), dorsal attention (two nodes), ventral attention (four nodes), fronto-parietal (four nodes), parietal memory (one node) and visual (five nodes) networks. Interestingly, these hub patterns were predominantly integrative networks.

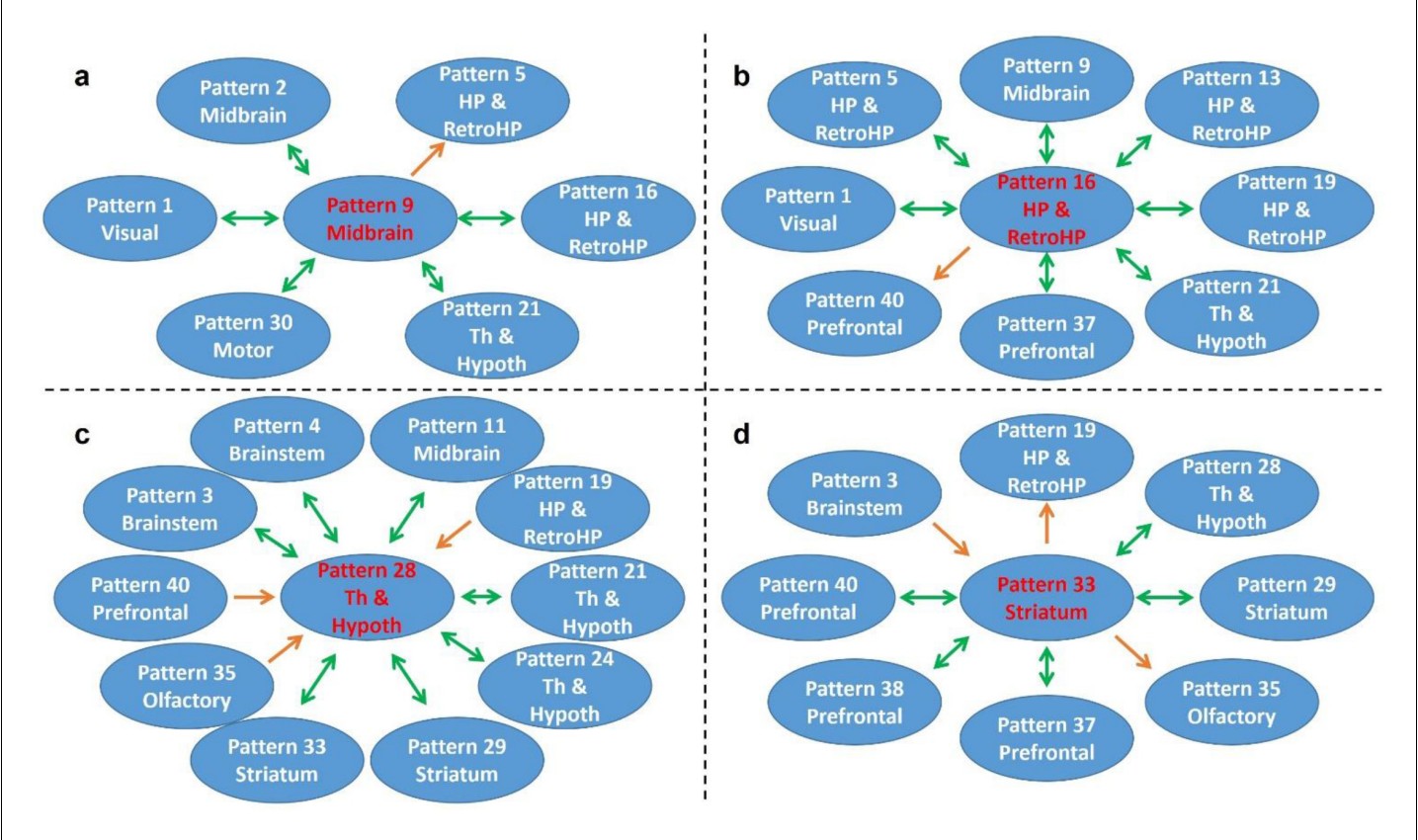

**Figure 7.** Transition patterns of hub networks. Green arrows denote bidirectional transitions between RSFC patterns. Orange arrows denote unidirectional transitions between RSFC patterns. Pattern numbers correspond to the seed map numbers shown in *Figure 2* and *Figure 2—figure supplement 1—5*. The brain system of the seed for each pattern is listed in the circle. Th and Hypoth, Thalamus and Hypothalamus; HP and RetroHP, Hippocampus and Retrohippocampus.

DOI: https://doi.org/10.7554/eLife.33562.016

## Discussion

In the present study, we investigated temporal sequential transitions between intrinsic brain activity patterns in the awake rat and human brain. We showed that transitions between RSFC patterns exhibited high reproducibility across animals and were significantly above chance (*Figures 3* and *4*). In addition, the RSFC pattern transition network was constructed using the thresholded transition matrix (*Figure 4*), and its topological organization including the community structure (*Figure 5*) and hubness (*Figure 6*) was evaluated. Moreover, the transitions of four representative hub RSFC patterns in rats were demonstrated (*Figure 7*). Importantly, non-random RSFC pattern transitions were also observed in humans (*Figure 8*), and the organization of the human transition network was further analyzed using the same graph analysis approach (*Figures 9* and *10*). Taken together, the present study for the first time characterized the temporal organization between successive brain connectivity configurations. It demonstrates that spontaneous brain activity was not only far from random spatially, but also far from random temporally. Similar results in rats and humans indicate that this feature might be well conserved across species. These data collectively have provided new insight into understanding the spatiotemporal dynamics of spontaneous activity in the mammalian brain.

### Method to unveil the temporal relationship between characteristic RSFC patterns

Although it has been well recognized that RSFC is dynamic in nature (*Hutchison et al., 2013a*), previous studies in this research line generally focused on revealing the spatial features of recurring

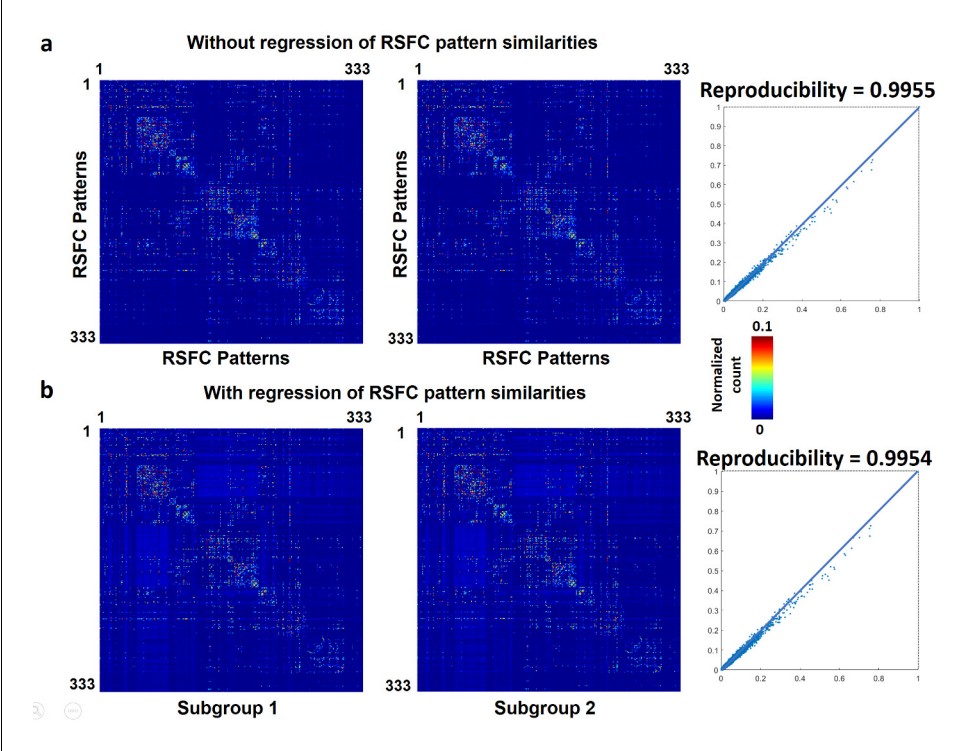

**Figure 8.** Reproducibility of temporal transitions between 333 characteristic RSFC patterns in humans. The transition matrices of two randomly divided subgroups (**a**) without regression of RSFC pattern similarities (**b**) with the regression of RSFC pattern similarities.
DOI: https://doi.org/10.7554/eLife.33562.017

RSFC patterns. Meanwhile, we have relatively sparse knowledge in the temporal relationship between characteristic RSFC patterns (*Majeed et al., 2011*; *Zalesky et al., 2014*; *Vidaurre et al., 2017*). To bridge this gap, we set out to systematically investigate temporal transitions between RSFC patterns.

To tackle this issue, we first need a set of representative RSFC patterns in the awake rat brain. Since the rat brain has ~6000 brain voxels at our spatial resolution ($0.5 \times 0.5 \times 1$ mm$^3$), in principle we can have ~6000 RSFC profiles in total. However, elucidating temporal sequences between such a large number of RSFC patterns is not only computationally intensive, but also unnecessary as many of these patterns are highly similar to each other. To obtain a survey of characteristic RSFC patterns, we adopted a RSFC-based parcellation of the awake rat brain (*Ma et al., 2016*). In this scheme, all ~6000 voxels were clustered into 40 parcels based on the similarity of their RSFC patterns, so that brain voxels' RSFC profiles were similar within each parcel but dissimilar across parcels (*Ma et al., 2016*). Notably, these parcels were highly reproducible between animals and exhibited high with-parcel homogeneity (*Ma et al., 2016*). Therefore, RSFC patterns obtained based on these parcels provided a comprehensive representation of all ~6000 RSFC patterns.

To examine the temporal relationship between these characteristic RSFC patterns, we adapted a recently developed method showing that BOLD co-activation patterns of rsfMRI frames well correspond to their instantaneous RSFC patterns (*Liu et al., 2013*; *Liu and Duyn, 2013*). This notion has been demonstrated in both humans, as well as in awake and anesthetized rats (*Liang et al., 2015*). Using this notion, each rsfMRI frame was corresponded to one of the 40 characteristic RSFC patterns based on the spatial similarity to the frame's BOLD co-activation pattern. The validity of this matching process was confirmed by high spatial similarity between averaged rsfMRI frames and matched characteristic RSFC patterns, quantified by their spatial correlations (*Figure 2* and *Figure 2—figure supplements 1–5*). This step resulted in a time sequence of RSFC patterns, which allowed us to systematically investigate the temporal transitions between these RSFC patterns.

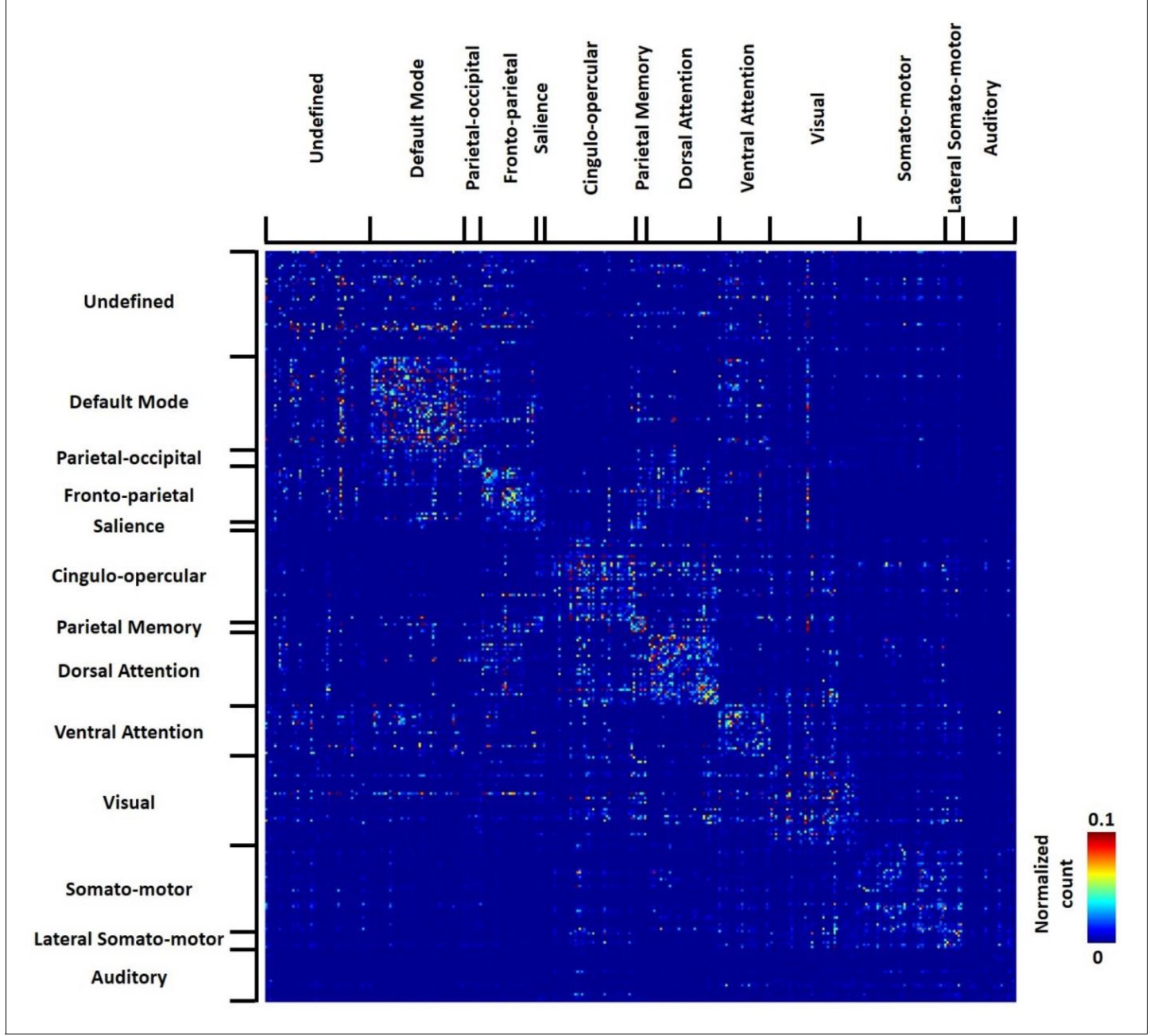

**Figure 9.** Thresholded group-level RSFC pattern transition matrix in humans (permutation test, p<0.05, FDR corrected). Rows/columns are grouped based on the brain system of the seed.

DOI: https://doi.org/10.7554/eLife.33562.018

## Nonrandom temporal transitions between RSFC patterns in rats

Our data showed that temporal transitions between RSFC patterns were highly reproducible in rats, reflected by significant reproducibility between randomly divided subgroups. In addition, these reproducible transitions were not dominated by a small portion of animals, evidenced by highly significant reproducibility at the individual level. To rule out the possible inflation of reproducibility resulting from the possibility that transitions between more similar RSFC patterns may occur at a higher chance in both subgroups, spatial similarities between characteristic RSFC patterns were regressed out in the transition matrices of both subgroups, and we found that the reproducibility of transitions remained high. These data show that transitions between RSFC patterns were robust and

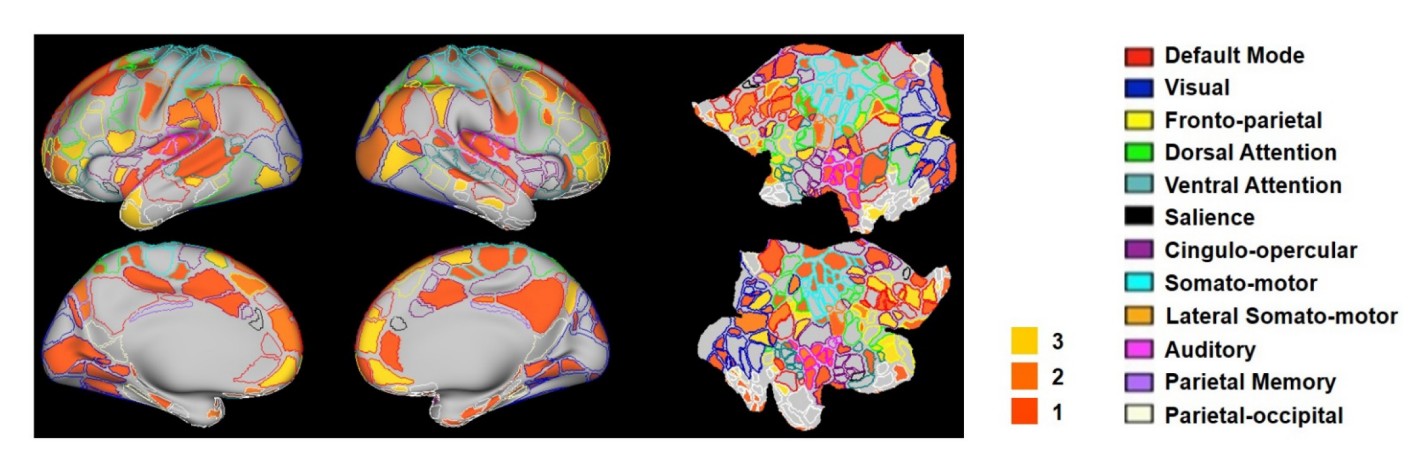

**Figure 10.** Hub scores of RSFC patterns in humans displayed on their seed regions. The boundary of nodes was color coded based on the brain system (right). Nodes with hub score ≥3 were defined as hubs (yellow-filled nodes).
DOI: https://doi.org/10.7554/eLife.33562.019

not dictated by RSFC pattern similarities. In addition, using permutation tests, we identified a number of transitions between RSFC patterns that were statistically above chance, further demonstrating that these transitions were non-random.

We also ruled out the possibility that RSFC pattern transitions were driven by head motion. Consistent transition matrices were obtained in two subgroups of animals with low (below median) and high (above median) motion levels (*Figure 3—figure supplement 1*). Using a permutation test, we confirmed that the reproducibility using motion-based division to subgroups was statistically not different from the reproducibility based on random division with (p=0.75) and without (p=0.67) regression of RSFC pattern similarities (*Figure 3—figure supplement 2*). In addition, no difference in head motion was observed between rsfMRI frames involved and not involved in a transition. We also found a minimal correlation between the mean head motion level for each transition sequence and the occurrence count of this transition sequence (r = −0.034), further indicating that RSFC pattern transitions were independent of head motion. Finally, consistent results were obtained at a more stringent motion censoring threshold (*Figure 3—figure supplement 3*), which demonstrated that our results were robust and insensitive to the motion censoring threshold selected.

Taken together, these data provide strong evidence indicating that RSFC patterns do not transit from/to each other in a random manner, but follow specific temporal sequences. This result well agrees with a recent report that spontaneous activity from ensembles of simultaneously recorded neurons was characterized by ongoing spatiotemporal activity patterns (*Mazzucato et al., 2015*), which recurred during all trials, and transitions between patterns could be reliably extracted using a hidden Markov model (*Mazzucato et al., 2015*).

## Transitions between RSFC patterns within and across brain systems in rats

We found that transitions between RSFC patterns occurred frequently between networks from the same brain system (*Figure 4*). This result might be attributed to the factor that seed regions of networks in the same brain system typically subserve similar brain function. In addition, regions in the same brain system are usually strongly connected with each other (*Liang et al., 2013*), and thus transitions between their RSFC patterns can frequently occur.

Our data also showed prominent cross-system transitions (*Figure 4*). For instance, switching between striatal networks and somatosensory/prefrontal cortical networks frequently occurred. Such cortical-subcortical system transitions might rely on the structural basis of corticostriatal projections identified in the rat brain (*Paxinos, 2015*). We speculate that bidirectional transitions between striatal and somatosensory/prefrontal RSFC networks might indicate the presence of both 'bottom-up' and 'top-down' processing involving high-order cortical and low-order subcortical regions at rest

(*Gurney et al., 2015*; *Piray et al., 2016*). In addition, significant transitions from striatal to thalamic/hippocampal RSFC networks indicate a close relationship between these subcortical systems, which can be further supported by strong RSFC between the CPu and thalamus found in the awake rat brain (*Liang et al., 2013*). Taken together, these results show non-trivial transitions between RSFC patterns within and across systems in the awake rat brain, and such transitions might play a critical role in coordinating spontaneous brain activity in separating brain systems.

## Organization of the RSFC pattern transition network in rats

A graph characterizing the transition network between RSFC patterns was constructed with each node representing a characteristic RSFC pattern and each edge denoting a statistically significant transition relationship between two nodes. We investigated the topological organization of this weighted directed graph including its community structure (*Figure 5*) and hubness (*Figure 6*). The transition network exhibited a prominent community structure evidenced by a high modularity, indicating that the global transition network between RSFC patterns was organized in a non-trivial manner.

We also identified several hub RSFC patterns (*Figure 6*) and scrutinized their transitions with other RSFC patterns (*Figure 7*). Hub patterns were central nodes in the RSFC transition network which played a pivotal role in transitions from/to other RSFC patterns. We found that the hippocampus RSFC network was pivotal to the transitions to the superior and inferior colliculi networks, as well as visual, prefrontal and orbital cortical networks. A recent study showed that low-frequency hippocampal–cortical activity drove brain-wide rsfMRI connectivity, highlighting the pivotal role of the hippocampus in RSFC transitions (*Chan et al., 2017*). In addition, Xiao and colleagues demonstrated that hippocampal spikes were associated with calcium cortical co-activation patterns in the visual and cingulate cortical regions (*Xiao et al., 2017*), consistent with our observation of transitions between hippocampal networks and visual/cingulate networks (*Figure 7*). Interestingly, it has also been reported that the hippocampus interacted with multiple cortical and subcortical regions in the form of sharp wave ripples (*Logothetis et al., 2012*), and the onset of such ripples was found to be controlled by the propagating signals from the cortex to hippocampus (*Mölle et al., 2006*; *Hahn et al., 2012*; *Roumis and Frank, 2015*). These results agree with the bidirectional transitions between hippocampal and cortical networks found in the present study (*Figure 7*). Although sharp wave ripples in the hippocampus and RSFC pattern transitions might be from different signal sources, the centralized role of the hippocampus is shared by these two forms of hippocampal-cortical information flow.

In accordance with our previous report that anterior ventral thalamus was a critical hub in the rat functional brain network (*Ma et al., 2016*), the RSFC pattern of anterior ventral thalamus was also a hub in the network of RSFC pattern transitions. A recent study investigating the relationship between single neuron spiking activity and brain-wide cortical calcium dynamics found that thalamic spikes could both predict and report (i.e. firing before and after) various types of large-scale cortical activity patterns, which were supported by slow calcium activities (<1 Hz) (*Xiao et al., 2017*). These data indicate a pivotal role of the thalamus in transitions between distinct spontaneous brain activity patterns. This result can be further supported by the finding that the ventral thalamus could recruit long-range cortical and subcortical networks and initiate their interactions through low-frequency (1 Hz) activity (*Leong et al., 2016*), and the thalamus could facilitate diverse and brain-wide functional neural integration in a specific spatiotemporal manner (*Liu et al., 2015*; *Leong et al., 2016*).

Further, our data revealed a hub of the RSFC pattern of the ventral CPu. As a part of the striatum, CPu is linked to multiple corticostriatal projections (*Paxinos, 2015*), and it might play a centralized role in transitions involving multiple cortical RSFC patterns (*Lee et al., 2017*). Taken together, these data indicate that hub RSFC patterns were central nodes linking multiple brain systems and might be critical for us to understand how activities from different brain systems are integrated to maintain normal brain function in rodents.

## RSFC pattern transitions in humans

To examine whether nonrandom RSFC pattern transitions we observed were only a specific feature in the rat brain, we investigated RSFC pattern transitions in humans by applying the same analysis approach to rsfMRI data from the HCP. We found that, like rats, transitions between RSFC patterns

were also nonrandom in humans, evidenced by highly consistent transition matrices between two randomly divided subject subgroups. This result well agrees with a recent study showing that dynamic switching between human brain networks was not random (*Vidaurre et al., 2017*). Interestingly, the split-group reproducibility was somewhat higher in humans than those observed in rats (both with and without regression of RSFC pattern similarities). This difference is likely due to much more human data used (406/406 human subjects v.s. 20/21 rats in each subgroup), which would average out larger amount of individual variability. This concept can be further supported by comparable reproducibility rates if we randomly picked 20 human subjects for each subgroup (reproducibility = 0.91 for human data v.s. reproducibility = 0.86 for rat data), as well as similar reproducibility at the individual level (human data: 0.6($\pm$0.05); rat data: 0.57 ($\pm$0.14)). Collectively, these results suggest that nonrandom transitions between characteristic RSFC patterns are not merely a specific feature in rodents, but conserved in both humans and rats.

The group-level RSFC pattern transition matrix was also thresholded using the permutation test. We found that, consistent with rat data, RSFC pattern transitions in humans more frequently occurred within the same brain system, but considerable cross-system transitions were also observed. We further calculated the hubness of individual nodes in the human RSFC transition network, and found multiple hubs belonging to separate brain systems including default-mode, cingulo-opercular, dorsal attention, ventral attention, fronto-parietal, parietal memory and visual networks. Intriguingly, virtually all hubs were integrative networks (with the only exception of the visual network) that are known to integrate information from multiple areas (e.g. sensori-motor systems). Our data suggest that these integrative networks are central in human RSFC pattern transitions. It has to be noted that direct comparisons of hubs between human and rat data is still premature as the human brain parcellation in Gordon et al.'s scheme did not include subcortical regions, while most transition hubs in the rat brain were subcortical networks. Such comparison is warranted in more detailed studies in the future. Nonetheless, these findings still highlight the translational utility of the analysis applied in the present study, which might shed light onto comparative neuroanatomy. Our results have also provided new insight into understanding the spatiotemporal dynamics of spontaneous activity in the mammalian brain.

## Potential limitation

One limitation of the present study is that single rsfMRI frames could exhibit features of more than one RSFC pattern. It should be noted that corresponding a rsfMRI frame to its most similar reference RSFC pattern is only an approximation for the purpose of investigating spatiotemporal dynamics of spontaneous brain activity. To mitigate this issue, we set a minimal threshold (correlation coefficient >0.1, $p<10^{-13}$) to remove rsfMRI frames that were not similar to any of 40 reference RSFC patterns (e.g. rsfMRI frames dominated by noise), and ensured that the similarity between each rsfMRI frame and the RSFC pattern it corresponded to was statistically significant after Bonferroni correction ($p<0.05/40834$ rsfMRI volumes $\approx 10^{-6}$). 89.9% of total rsfMRI volumes met this criterion, indicating that reference RSFC patterns indeed captured most spontaneous brain activity patterns in the awake rat brain.

## Conclusions

In conclusion, the present study investigated temporal transitions between spontaneous brain activity patterns in the awake rat and human brain. We found that these transitions were far from random in both species, demonstrating that this feature might be a general phenomenon in the mammalian brain. Using graph theory analysis, our study further revealed central RSFC patterns in the transition networks. This study has opened a new avenue to investigating the spatiotemporal organization of spontaneous activity in the mammalian brain.

## Materials and methods

### Animals

41 Long-Evans (LE) adult male rats were used. Data from 31 rats were also used in another study (*Ma et al., 2016*) and were reanalyzed for the purpose of the present study. All rats were housed in Plexiglas cages with controlled ambient temperature (22–24°C) and maintained on a 12 hr light:12 hr

dark schedule. Food and water were provided ad libitum. The experiment was approved by the Institutional Animal Care and Use Committee (IACUC) at the Pennsylvania State University.

## Rat MRI experiments

Rats were acclimated to the MRI environment for seven days following the procedures described in (*Zhang et al., 2010*; *Liang et al., 2011*, *2012a*, *2012b*, *2014*; *Gao et al., 2017*) to minimize motion and stress. For the setup of awake animal imaging, the rat was first briefly anesthetized with 2–3% isoflurane and fit into a head holder with a built-in coil and a body tube. Isoflurane was then discontinued and the rat was placed into the magnet. All rats were fully awake during imaging. We measured the respiratory rate in a separate cohort of animals (n = 16) that were imaged using the same setting. The mean (±SD) breathing rate = 86.2 (±15.5), which was well within the normal range of breathing rate in rats (70–100 Hz). This measurement was also consistent with the normal physiological state measured in rats acclimated to MRI environment using similar methods in other labs (*King et al., 2005*; *Ferenczi et al., 2016*), suggesting that animals were well adapted to the MRI environment during imaging. A similar approach has also been used for awake rodent fMRI in other groups (*Bergmann et al., 2016*; *Chang et al., 2016b*; *Yoshida et al., 2016*).

MRI data acquisition was conducted on a Bruker 7T small animal MRI scanner (Billerica, MA). Anatomical MRI images were acquired using a T1-weighted rapid imaging with refocused echoes (RARE) sequence with the following parameters: repetition time (TR) = 1500 ms; echo time (TE) = 8 ms; matrix size = 256 × 256; field of view (FOV) = 3.2 × 3.2 cm$^2$; slice number = 20; slice thickness = 1 mm; RARE factor = 8. rsfMRI images were acquired using a T2*-weighted gradient-echo echo planar imaging (EPI) sequence with the following parameters: TR = 1000 ms; TE = 15 ms; matrix size = 64 × 64; FOV = 3.2 × 3.2 cm$^2$; slice number = 20; slice thickness = 1 mm. 600 EPI volumes were acquired for each run, and two to four runs were acquired for each animal.

## Rat image preprocessing

Detailed description of the image preprocessing pipeline can be found in (*Ma et al., 2016*) and is briefly summarized as follows. Relative FD (*Power et al., 2012*) of rat brain EPI images was calculated, and EPI volumes with FD >0.2 mm and their immediate temporal neighbors were removed (1.75% of total rsfMRI volumes). The first 10 volumes of each rsfMRI run were also removed to warrant a steady state of magnetization. Brain normalization to a standard rat brain was performed using Medical Image Visualization and Analysis (MIVA, http://ccni.wpi.edu/). Head motion was corrected using SPM12 (http://www.fil.ion.ucl.ac.uk/spm/). In-plane spatial smoothing was carried out using a Gaussian filter (FWHM = 0.75 mm). Nuisance regression was performed with the regressors of three translation and three rotation motion parameters estimated by SPM as well as white matter and ventricle signals. Band-pass filtering was performed with the frequency band of 0.01–0.1 Hz.

## Characteristic RSFC patterns

To obtain a library of characteristic RSFC spatial patterns in the awake rat brain, we used a RSFC-based whole-brain parcellation scheme (40 non-overlap parcels) we previously published (*Ma et al., 2016*). In this scheme, voxels with similar RSFC patterns were grouped together, so that RSFC patterns were similar within parcels but dissimilar across parcels (*Ma et al., 2016*). As a result, these 40 RSFC patterns represented a set of characteristic RSFC patterns in the awake rat brain and were used as the references (also see Supplemental Information).

All characteristic RSFC patterns were obtained using seed-based correlational analysis with each parcel as the seed. Specifically, the regionally-averaged time course from all voxels within the seed region was used as the seed time course, and the Pearson cross-correlation coefficient between the seed time course and the time course of each individual brain voxel was calculated. Correlation analysis was performed for the first 540 volumes of each rsfMRI run to ensure the same degree of freedom. Correlation coefficients were then Fisher's Z-transformed. For each parcel, its group-level RSFC map was voxelwise calculated using one-sample t-test based on a linear mixed-effect model with the random effect of rats and the fixed effect of Z values for each run. The spatial similarity between these reference RSFC patterns was determined by pairwise spatial correlations between every two characteristic RSFC patterns.

## Temporal transitions between RSFC patterns

To analyze temporal transitions between RSFC patterns, a time sequence of framewise RSFC patterns (1 s each frame) was first obtained by matching each rsfMRI frame to one of the 40 reference RSFC patterns, based on the notion that BOLD co-activation patterns in single rsfMRI frames also represent their RSFC patterns (*Liu et al., 2013*; *Liu and Duyn, 2013*; *Liang et al., 2015*). To do so, preprocessed rsfMRI time series were first demeaned and variance normalized. Subsequently, the spatial Pearson correlation coefficients between each rsfMRI frame and individual reference RSFC patterns in the library were respectively calculated. The reference RSFC pattern that best matched the rsfMRI frame (i.e. the reference RSFC pattern that had the highest spatial correlation) was selected. To ensure the correspondence between each rsfMRI frame and the matched RSFC pattern was statistically meaningful, we set a minimal threshold of the spatial correlation coefficient >0.1 (p value < $10^{-13}$). 89.9% of total volumes met this criterion. Frames that did not meet this criterion (10.09% of total volumes) were labeled as not corresponding to any reference RSFC patterns. This step generated a time sequence of framewise RSFC patterns. In this sequence, each rsfMRI frame was denoted by a number between 1 and 40, representing its correspondence to one of the 40 reference RSFC patterns. The number 0 was used to denote rsfMRI frames not corresponding to any reference RSFC patterns, as well as frames removed in image preprocessing (e.g. frames with large FD). In the sequence, the number of transitions between every two RSFC patterns was counted (i - > j, where i ≠ j, i ≠ 0 and j ≠ 0). Transitions involving 0 (i.e., 0 - > 0, or 0 - > i, or i - > 0, where i ≠ 0) were not counted. This procedure yielded a 40 × 40 RSFC pattern transition matrix, where its entry (i, j) represented the number of transitions between RSFC pattern i to pattern j.

## Reproducibility of temporal transitions between RSFC patterns

The reproducibility of temporal transitions between RSFC patterns was assessed at both the group and individual levels. At the group level, we used a split-group approach. All 41 rats were randomly divided into two subgroups with 20 rats in subgroup 1 and 21 rats in subgroup 2. The RSFC pattern transition matrix was computed for each subgroup. Entries in each matrix were normalized to the range of [0, 1], and the correlation of the corresponding off-diagonal matrix entries between the two subgroups was assessed.

It is possible that spatially similar RSFC patterns had a higher chance to transit between each other in both subgroups, and this systematic bias might inflate the reproducibility of RSFC pattern transitions between the two subgroups. To control for this effect, we regressed out the spatial similarity between every two reference RSFC patterns, quantified by their spatial correlation value, from the transition matrices in both subgroups and then assessed the reproducibility again.

Reproducibility of temporal transitions between RSFC patterns was also evaluated at the individual level. For each rat, its individual-level transition matrix was obtained, and the reproducibility was computed using Pearson correlation of the corresponding off-diagonal matrix entries between this individual-level transition matrix and the group-level transition matrix.

## Organization of RSFC pattern transitions

The group-level transition matrix was thresholded to identify transitions that were statistically significant. The p value of each entry in the transition matrix was calculated using the permutation test. Since we were only interested in transitions between two different RSFC patterns, before the permutation test, the temporal sequence of RSFC patterns was consolidated by combining consecutively repeated appearances of the same pattern to one appearance of the pattern. For example, four consecutive appearances of Pattern 'x' (i.e. 'xxxx') were replaced by one 'x'. This consolidated temporal sequence was then permuted 10000 times, and a transition matrix was obtained for each permuted sequence. This step generated an empirical null distribution for each off-diagonal entry in the transition matrix, and the p value of the entry was obtained accordingly. p values were further adjusted using false-discovery rate (FDR) correction at the rate of 0.05 (*Genovese et al., 2002*). Entries with insignificant p values were set to zero. All entries were then rescaled to the range of [0,1]. Finally, similarities between RSFC patterns were regressed out from nonzero entries.

Using this thresholded transition matrix as the adjacency matrix, a graph was constructed using Gephi 0.9.1 (https://gephi.org/). In this weighted directed graph, each node represented a RSFC

pattern, and each edge connecting two nodes signified an above-chance transition between two RSFC patterns with the edge weight proportional to the normalized number of transitions.

Graph theory analysis of this RSFC pattern transition network was performed using the Brain Connectivity Toolbox (https://sites.google.com/site/bctnet/). The community affiliation of nodes in the graph was obtained by repeating the Louvain community detection algorithm (*Vincent et al., 2008*) for 1000 times to ensure a stable solution. Specifically, for each repetition, a $40 \times 40$ matrix was generated so that its entry (i, j) was one if nodes i and j were in the same community and 0 otherwise. The average of these 1000 matrices was then binarized using a threshold of 0.9, and the final inference of the community affiliation was obtained from the node affiliation of connected components in the binarized matrix (*Liang et al., 2011*).

To identify the hub nodes in the transition graph, local graph measures of node strength, betweenness centrality, local characteristic path length and local clustering coefficient of each node were first computed. Using these node metrics, hub nodes with high node strength, high betweenness centrality, short distance to other nodes, and low local clustering coefficient (*Bullmore and Sporns, 2009*) were identified using the method described in (*van den Heuvel et al., 2010*). Briefly, a hub score (0 to 4) was given to each node according to the total number of the following criteria the node met: (1) upper 20 percentile in node strength; (2) upper 20 percentile in betweenness centrality; (3) lower 20 percentile in characteristic path length; and (4) lower 20 percentile in local clustering coefficient. Node met at least three criteria was defined as a hub (i.e. hub score $\geq$3), indicating its pivotal role in transitions between RSFC patterns.

## Reproducibility of RSFC pattern temporal transitions in the human brain

The reproducibility of temporal transitions between RSFC patterns in the human brain was evaluated using a similar process. The human data used were the 'extensively preprocessed 3T rsfMRI data' from 812 subjects, which were a subset of the S1200 Subjects Data Release of the Human Connectome Project (HCP, https://www.humanconnectome.org/) (*Van Essen et al., 2013*). All rsfMRI data were acquired on a 3T Siemens Skyra MRI scanner using a multi-band EPI sequence with the parameters of TR = 720 ms, TE = 33.1 ms, flip angle = 52°, FOV = $208 \times 180$ mm$^2$, matrix size = $104 \times 90$, voxel size = $2 \times 2 \times 2$ mm$^3$, slice number = 72, slice thickness = 2 mm, multiband factor = 8 (*Feinberg et al., 2010*; *Moeller et al., 2010*; *Setsompop et al., 2012*; *Glasser et al., 2013*). Data preprocessing used the HCP minimal preprocessing pipelines (*Glasser et al., 2013*), MSM-All brain registration (*Robinson et al., 2014*). Head motion correction was conducted using the ICA + FIX pipeline (*Griffanti et al., 2014*; *Salimi-Khorshidi et al., 2014*), and these procedures were completed by the HCP.

To obtain a library of characteristic RSFC patterns in the human brain, we used a well-established RSFC-based parcellation scheme (333 parcels) (*Gordon et al., 2016*), which has been demonstrated to have high within-parcel homogeneity and reflect the underlying connectivity structure of the human brain (*Gordon et al., 2016*). Based on this scheme, 333 characteristic RSFC patterns were obtained using seed-based correlational analysis with individual parcels as seeds. For each rsfMRI run, the seed time course was averaged from all grayordinates within the seed, and Pearson cross-correlation coefficient between the seed time course and the time course of each individual cortical grayordinate was calculated. Correlation coefficients were Fisher's Z-transformed. For each parcel, its group-level RSFC map was grayordinate-wise calculated by one-sample t-test using a linear mixed-effect model with the random effect of subjects and the fixed effect of Z values for individual runs. Pairwise spatial correlations between these group-level RSFC maps were also calculated to measure their similarities.

Each rsfMRI frame was matched to one of the 333 reference patterns that had the highest spatial similarity to the BOLD co-activation pattern of the frame, gauged by their spatial Pearson correlation. The minimal spatial correlation coefficient was set at 0.05. rsfMRI frames below this threshold were labeled as not corresponding to any reference RSFC patterns. This step generated a temporal sequence of RSFC patterns for each rsfMRI run. In this sequence, each rsfMRI frame was denoted by a number between 1 and 333, representing its correspondence to one of the 333 characteristic RSFC patterns. The number 0 was used to denote rsfMRI frames not corresponding to any reference RSFC patterns. In the sequence, the number of transitions between every two RSFC patterns was counted (i - > j, where i $\neq$ j, i $\neq$ 0 and j $\neq$ 0). Transitions involving 0 (i.e., 0 - > 0, or 0 - > i, or i - > 0,

where i $\neq$ 0) were not counted. This procedure yielded a 333 $\times$ 333 temporal transition matrix for each run. The temporal transition matrix for each subject was obtained by summing the temporal transition matrices from all four rsfMRI runs. The group-level transition matrix was obtained by averaging the subject-level temporal transition matrices across all subjects.

The reproducibility of temporal transitions between RSFC patterns in the human brain was also assessed at both the individual and group levels. The individual-level reproducibility was calculated based on the correlation of off-diagonal entries between the individual's transition matrix and the group-level transition matrix. The group-level reproducibility was evaluated using a split-group approach. All 812 subjects were randomly divided into two subgroups (406 subjects each subgroup). The RSFC pattern transition matrix was computed for each subgroup, respectively, and the correlation of off-diagonal matrix entries between the two subgroups were assessed. To control for the effect of the spatial similarity between characteristic RSFC patterns on the reproducibility measure, we also regressed out the spatial correlation values between characteristic RSFC patterns from the transition matrices in both subgroups and then re-assessed the reproducibility. All reproducibility assessment was based on unthresholded matrices.

To construct the human RSFC pattern transition network, the group-level RSFC transition matrix obtained from all 812 subjects was thresholded using same permutation test as that used in rats, and the graph theory analysis was applied in a similar manner.

## Acknowledgements

The present study was partially supported by National Institute of Neurological Disorders and Stroke Grant R01NS085200 (PI: Nanyin Zhang, PhD) and National Institute of Mental Health Grant R01MH098003 and RF1MH114224 (PI: Nanyin Zhang, PhD). Part of this research was conducted using the high-performance computing resources provided by the Institute for CyberScience at the Pennsylvania State University (https://ics.psu.edu). Human data were provided by the HCP, WU-Minn Consortium (Principal Investigators: David Van Essen and Kamil Ugurbil; 1U54MH091657) funded by 16 NIH Institutes and Centers that support the NIH Blueprint for Neuroscience Research; and by the McDonnell Center for Systems Neuroscience at Washington University.

## Additional information

### Funding

| Funder | Grant reference number | Author |
| --- | --- | --- |
| National Institute of Mental Health | R01MH098003 | Nanyin Zhang |
| National Institute of Neurological Disorders and Stroke | R01NS085200 | Nanyin Zhang |
| National Institute of Mental Health | RF1MH114224 | Nanyin Zhang |

The funders had no role in study design, data collection and interpretation, or the decision to submit the work for publication.

### Author contributions

Zhiwei Ma, Software, Formal analysis, Validation, Visualization, Writing—original draft; Nanyin Zhang, Conceptualization, Formal analysis, Supervision, Funding acquisition, Validation, Investigation, Visualization, Methodology, Writing—original draft, Project administration, Writing—review and editing

### Author ORCIDs

Nanyin Zhang http://orcid.org/0000-0002-5824-9058

## Ethics

Human subjects: This study only involves analysis of human imaging data that were publicly available (Human Connectome Project). No informed consent was obtained. The study has been approved by the IRB of the Pennsylvania State University (STUDY00005665).

Animal experimentation: This study was performed in strict accordance with the recommendations in the Guide for the Care and Use of Laboratory Animals of the National Institutes of Health. All of the animals were handled according to approved institutional animal care and use committee (IACUC) protocols (#43583-1) of the Pennsylvania State University.

## Decision letter and Author response

Decision letter https://doi.org/10.7554/eLife.33562.024
Author response https://doi.org/10.7554/eLife.33562.025

## Additional files

### Supplementary files

• Transparent reporting form
DOI: https://doi.org/10.7554/eLife.33562.020

### Major datasets

The following previously published dataset was used:

| Author(s) | Year | Dataset title | Dataset URL | Database, license, and accessibility information |
|---|---|---|---|---|
| Van Essen DC, Smith SM, Barch DM, Behrens TE, Yacoub E, Ugurbil K | 2013 | The Human Connectome Project S1200 Data Release | https://www.humanconnectome.org/study/hcp-young-adult/document/1200-subjects-data-release | Open access dataset available from ConnectomeDB (https://db.humanconnectome.org/app/template/Login.vm). Account registration is required and access to certain data elements such as family structure is subject to restricted use terms (please see http://www.humanconnectome.org/study/hcp-young-adult/data-use-terms). |

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
