## [Decision Letter]

[Editors’ note: a previous version of this study was rejected after peer review, but the authors submitted for reconsideration. The first decision letter after peer review is shown below.]

Thank you for submitting your work entitled "Temporal transitions of spontaneous brain activity in awake rats" for consideration by *eLife*. Your article has been evaluated by a Senior Editor and three reviewers, one of whom, Jan-Marino Ramirez (Reviewer #1), is a member of our Board of Reviewing Editors. ion have agreed to reveal their identity:

Our decision has been reached after consultation between the reviewers. Based on these discussions and the individual reviews below, we regret to inform you that your work will not be considered further for publication in *eLife*.

The reviewers found the topic and phenomenon described very interesting and significant, but they raised several methodological concerns that would require considerable time to address. Also, the study is an extension of a prior study, and thus conceptually not sufficiently novel. We therefore believe it is better suited for a more specialized Journal.

Reviewer #1:

The discovery that the BOLD signal can be used to measure the resting state functional connectivity had a major impact on our understanding of brain functions and the development of the central nervous system. This non-invasive imaging approach allows the characterization of brain connectivity in humans and animal models. The signal is generally considered to be stationary, and provides insights into the temporal correlations of this so-called default state between different brain regions. Indeed, the assumption of a stationary nature is critical since any changes in these correlations are assumed to be due to differences in functional connectivity. The identification of temporal variability is therefore very significant. The present study provides evidence for a temporal variability. It is not the first study to do so, hence the study is not entirely novel, and the authors need to better emphasize what sets this study apart from other studies (including their own publication).

There are two issues that somewhat dampen my enthusiasm:

1) Given that a similar finding was already reported in an animal study, the addition of human MRI studies using the same approach would significantly enhance the impact of this study. To the best of my knowledge this has not been demonstrated in humans.

2) To what extent can the authors exclude movement artifacts. Given the important implications of these temporal shifts, it is important to exclude movements as a cause. Also can the authors identify any other reproducible reasons for the temporal variability?

Reviewer #2:

The manuscript by Ma et al. detailed the investigation of the transitions of rsfMRI functional connectivity (RSFC) network patterns in an awake rat. The authors' method of analysis that combines several recent approaches from prior publications – parcellation of RSFC patterns in an awake animal and utilizing graph theory approaches to identify) – is innovative. At present, the findings are novel and such analysis method could potentially be utilized to reveal similar dynamics in human rsfMRI studies. However, the manuscript can be significantly strengthened by further analyses of the current data and the discussions/interpretations improved by better integrating several pieces of prior literature that are presently omitted. I believe that English editing would also further improve this manuscript as there are grammatical errors and at places, the message is not completely clear.

1) The authors stated that each rsfMRI run, which comprises of 600 volumes, was divided into rsfMRI frames that were then statistically matched to the 40 characteristic RSFC patterns. The authors then made a comment that 89.9% of the total rsfMRI volumes met the minimum threshold of the spatial correlation coefficient > 0.1. I have two questions regarding these statements:

a) How long was each rsfMRI frame? I was unable to find the details in the Materials and methods section.

b) How well does each rsfMRI frame correspond to one of the 40 characteristic RSFC patterns? It would be best if the authors show and compare the representative 40 RSFC patterns generated from rsfMRI frames and those that were generated from a rsfMRI run (as shown in Figure 2 and Figure 2—figure supplements 1-39). This is because the number of time points included within each rsfMRI run will affect the sensitivity in detecting RSFC patterns and likely influence the subsequent analysis.

2) It is difficult to understand and interpret Figure 5, I believe its quality can be significantly improved. The authors should improve the visualization of the transition network as depicted in Figure 5A and how that translates to Figure 5B. As *eLife* is a general life and biomedical sciences journal, it would be difficult for readers to grasp the concept of such a complex graph theoretical network analysis just by reading the details in the Materials and methods section. The figure here should be used to best portray a simplistic yet informative summary of the analysis technique and the scientific findings.

3) It is still unclear what is the biological significance in the detection of numerous types of transition RSFC patterns. The authors only describe the observation of such transitions but based on the present discussions, it is unclear how valuable or valid these transition patterns are. Moreover, the current discussions made by the authors are insufficient, superficial and sometimes made without citations. For example, the authors state that "bidirectional transitions between striatal and somatosensory/prefrontal RSFC might indicate the presence of bottom-up and top-down processing…" without citing a source. At present, there are numerous studies that interrogated the large-scale effects of optogenetically stimulating striatal, hippocampal and thalamic neurons (Chan et al., 2017; Lee et al., 2016; Leong et al., 2016; Liu et al., 2015; Schmitt et al., 2017). In fact, the authors also characterized the effects of medial prefrontal cortex stimulation (Liang et al., 2015). As such, the authors should make an effort to integrate these pieces of literature or any other relevant works. Note that the literatures suggested here are not exhaustive and that the authors should expand their scope of discussion to better encompass and integrate literature in the interpretation of their findings.

4) Recently, a paper by Xiao et. al investigated the relationship between single neuron spiking activity and brain-wide cortical calcium dynamics (Xiao et al., 2017). They found that thalamic spikes could produce various types of large-scale cortical activity patterns, which are supported by large-scale slow calcium activities (<1 Hz). More importantly, they demonstrated slow calcium cortical activation patterns in the visual cortical and cingulate regions generated from hippocampal spikes. This finding seems relevant to the observations in the present study as the authors also found the transition between hippocampal networks and visual/cingulate networks. Moreover, Logothetis et al. also showed diffuse responses to hippocampal sharp wave ripple events in multiple cortical areas (e.g., visual and cingulate cortices) (Logothetis et al., 2012). With that in mind, how does the present data fit into the current literature landscape?

5) In the Discussion subsection “Nonrandom temporal transitions between RSFC patterns”, the authors state that the results of the present study were consistent with previous studies showing that BOLD signal periodically propagated from lateral to medial cerebral cortex (Majeed et al., 2011; Majeed et al., 2009). This statement seems out of place and erroneous as the findings of the present study did not demonstrate any propagation characteristics. Furthermore, the preceding statements in the same Discussion section addressed the reproducibility of the transitions between RSFC patterns, which is out of the context of the works by Majeed et al.

Reviewer #3:

The current report examines the spatiotemporal dynamics of spontaneous brain activity in the awake rat, examining whether transitions between RSFC states are nonrandom. While the paper is timely and of potential interest, the overall novelty is limited given previous reports by the same authors (Liang et al., 2015 Ma et al., 2016) and the restricted scope of the analyses conducted.

1) Imaging awake rats is advantageous (closer to human RSFC, eliminates effects of anesthesia, etc.), but is more sensitive to head motion. Laumann et al. (2016, Cerebral Cortex) have previously shown that part of the observed dynamics during rest is attributable to head motion. Since motion denoising in this study is lenient (FD exclusion criterion of 0.2 mm as is commonly used in humans) and no global signal regression was applied (which eliminates motion-related artifacts more effectively than ventricle and white matter signal regression alone), it is possible that the non-random transitions between RSFC states are due to head motion. The authors should quantify whether motion events trigger specific transitions. In addition, to more conclusively demonstrate that the results are independent of head motion, the authors can conduct the reproducibility analysis on two sub-groups based on head motion (below an above median FD).

2) The finding of non-random transition is novel and will be of interest to the extended community using RSFC to understand functional organization. That said, many aspects of this study are similar to previous studies. Most prominently: RSFC dynamics in awake rats were reported by the authors (Liang et al., 2015). However, the authors do not discuss the relations between the CAP analysis in Liang et al. to the current findings. In addition, the hub analysis replicates some findings from stationary RSFC analysis (Ma et al., 2016), but some findings differ (areas that appear in only one of the analyses). The authors should elaborate on this issue and explain the differences.

3) A significant shortcoming of this work is relating it to human RSFC. The manuscript would have been much stronger were the authors able to demonstrate that the observed transitions are present in humans and compare the similarities and differences in the dynamics of RSFC.

4) In the Discussion, the authors conclude that RSFC patterns obtained based on these parcels are characteristic and provide a comprehensive representation [to] all ~6000 RSFC patterns. This conclusion goes beyond the data. The number of 40 parcels was chosen by the authors in their previous report (Ma et al., 2016) as an example for a low-dimensionality parcellation, it's possible the high dimensionality parcellation will better represent the ~6000 voxels of the rat brain, especially when taking into account the fact that the rat brain is known to have more than 40 regions. To support their conclusion, the authors need to demonstrate that the high-dimensionality parcellation from Ma et al., 2016 (130 parcels) does not overperform the 40 parcels.

5) The authors fail to cite many recent publications relevant to the current work, including awake rodent fMRI from other groups (e.g., Bergmann et al., 2016; Chang et al., 2016; Yoshida et al., 2016), dynamic RSFC in mice (Grandjean et al., 2017) and spatiotemporal dynamics of intrinsic signals (Mohajerani et al., 2013).

[Editors’ note: what now follows is the decision letter after the authors submitted for further consideration.]

Thank you for submitting your article "Temporal transitions of spontaneous brain activity" for consideration by *eLife*. Your article has been favorably evaluated by Richard Ivry (Senior Editor) and three reviewers, one of whom, Jan-Marino Ramirez (Reviewer #1), is a member of our Board of Reviewing Editors. The following individual involved in review of your submission has agreed to reveal their identity: Ed Wu (Reviewer #2).

The reviewers have discussed the reviews with one another and the Reviewing Editor has drafted this decision to help you prepare a revised submission.

Summary:

This manuscript characterizes the transitions of resting-state functional MRI (rsfMRI) functional connectivity (RSFC) network patterns in awake rats as well as humans. The authors use innovative analysis methods including the parcellation of RSFC patterns in an awake animal and the utilization of graph theory approaches. The authors have addressed many of the issues that were raised previously, but we suggest additional revisions that will further improve this interesting study.

Essential revisions:

1) The addition of human data is important and increases the impact of this study. However, the high correlation values between the 2 subgroups (>0.995) and the fact that significant transitions are very sparse, bring to question the role of statistical thresholding. Since only a small fraction of the transitions are significant, thresholding in-and-of-itself can artificially increase correlation values between subgroups. In addition, thresholding can explain the higher reproducibility in humans relative to rats (human matrices seem sparser). The authors should examine the matrices with all of the raw data (i.e., without applying a threshold).

2) Along the same lines, in order to draw strong similarities/parallels between the human and rat rsfMRI data, the authors included the analysis outcome of the reproducibility matrix of RSFC patterns in humans (Figure 8) to demonstrate that the transition of RSFC patterns are nonrandom and echo the observations found in the awake rat brain (Figure 3). However, the authors should also include additional analysis outcomes that were generated from the human rsfMRI dataset as outlined by the analysis flowchart in Figure 1, such as RSFC pattern transition matrix (Figure 4) and graph theoretical analyses (Figures 5 and 6). It would appear from the flowchart that most of these analyses were already performed in order to generate the reproducibility matrix (Figure 8). Only then, a stronger case could be made by comparing multi-dimensional data to draw similarities between the rat and human brain, particularly in RSFC pattern transitions across time. On a minor note, it may also be of interest to look at the parcellated regions in the human brain using the method developed by the authors as there are numerous recent papers that attempt to improve parcellation procedures (Glasser et al., 2016).

3) The possibility that motion-related artifacts may explain some of the data continues to be a major potential concern, and the authors need to address this concern in a systematic manner.

3a) Based on the relatively modest difference between random and motion-based splitting (0.77 vs. 0.786) the authors claim that the motion-based reproducibility analysis suggested indicates that the high reproducibility in RSFC pattern transitions was not due to head motion. However, this inference is not supported by a statistical test. A permutation test in which the authors calculate reproducibility for many random divisions to subgroups and compare the distribution of correlations would allow this type of inference. In addition, the authors described the results of this analysis only after the regression of seed map similarities, but the supplementary figures should replicate the full Figure 3 and show the results both with and without this regression. Moreover, the authors should describe the FD values (mean+SD) of the subgroups above and below the median FD so the reader will be able to evaluate the data quality.

3b) In their second analysis, the authors examine whether transitions predict head motion. However, this approach is not very sensitive since it tests whether all transitions are caused by motion and not whether motion causes some of the transitions. The authors should test whether transitions are more likely to follow head motion and if such transitions are random across areas or affect specific areas preferentially.

3c) An alternative approach to address this issue is lowering the motion censoring threshold. The current one (0.2 mm) is taken from the human fMRI literature, and might not be suitable for the rat brain. Similar results with FD thresholds of 0.15 mm, 0.1 mm and 0.05mm can indicate that the transitions are motion independent.

4) Along the same lines, no description in the motion scrubbing methods is given for the human data. Did the authors exclude any motion-related artifacts?

5) The amount of raw data presented is considerable, but the number of second-level quantitative analyses is limited. This shortcoming makes the manuscript difficult to read and not very attractive for a broad audience. Below are some suggestions to address this concern:

5a) In Figure 2, the addition of the distribution of correlation between Seed maps and Average of rsfMRI frames for the 40 seeds will be highly advantageous. Currently, the reader has to visually evaluate 40 seeds in Figure 2—figure supplement 1. Note that parts of Figure 2—figure supplement 1 are cropped (left and lower aspects).

5b) In Figure 4, the authors present a 40x40 matrix. It is impossible to properly evaluate it. Others have used grouping to categories (based on the axes / brain systems) to assist the reader in evaluating which brain systems are characterized by transitions.

5c) Figure 6 depict seed maps that were already presented in previous figures. This redundancy is unwarranted. The authors can plot the hub score of the different 40 seeds and annotate brain systems.

5d) Figure 7 depicts, again, seed maps that were previously shown. The authors need to find a more compact presentation.

5e) Figure 8 depicts the human matrix. If the rat 40x40 matrix was challenging to evaluate, the human 333x333 matrix is virtually unreadable. The authors need to find a visual representation proper for this data via alternative visualization methods (perhaps a scatterplot will be a better way to compare between the 2 subgroups).

6) The authors need to better explain the significance/innovation of the present study, given that previous publications from the authors' laboratory and others have already demonstrated characteristics of RSFC pattern transitions across time (Chang et al., 2016; Liang et al., 2015; Liu and Duyn, 2013; Liu et al., 2015). It is clear that these already published studies did not extensively or exclusively document the transition of multiple RSFC patterns across time, but, the authors need to better explain why the present study is not just an incremental advance.

---

## [Author Response]

[Editors’ note: the author responses to the first round of peer review follow.]

Reviewer #1:The discovery that the BOLD signal can be used to measure the resting state functional connectivity had a major impact on our understanding of brain functions and the development of the central nervous system. This non-invasive imaging approach allows the characterization of brain connectivity in humans and animal models. The signal is generally considered to be stationary, and provides insights into the temporal correlations of this so-called default state between different brain regions. Indeed, the assumption of a stationary nature is critical since any changes in these correlations are assumed to be due to differences in functional connectivity. The identification of temporal variability is therefore very significant. The present study provides evidence for a temporal variability. It is not the first study to do so, hence the study is not entirely novel, and the authors need to better emphasize what sets this study apart from other studies (including their own publication).

We appreciate the general comment. Regarding the novelty, the present study for the first time characterized the *temporal sequences* between brain connectivity patterns and showed that spontaneous brain activity did not fluctuate randomly, but followed specific orders. Our study is different from previous dynamic rsfMRI studies, which mainly focused on the spatial characteristics of RSFC dynamics. We believe our results provide new insight into comprehensively characterizing spatiotemporal dynamics of spontaneous brain activity.

There are two issues that somewhat dampen my enthusiasm:1) Given that a similar finding was already reported in an animal study, the addition of human MRI studies using the same approach would significantly enhance the impact of this study. To the best of my knowledge this has not been demonstrated in humans.

We agree with the reviewer that temporal transitions of brain connectivity patterns have not been studied in humans, and the addition of human rsMRI studies using the same approach would significantly enhance the impact of this study. To address this issue, in the revised manuscript we have included new results based on human rsfMRI data collected from 812 subjects in the Human Connectome Project (HCP). Briefly, we found that, like our results in rats, transitions between brain connectivity patterns were highly reproducible in humans, suggesting that transitions between characteristic resting-state functional connectivity (RSFC) patterns are nonrandom. These data demonstrated that this feature was well conserved across species in rats and humans, and might represent a general phenomenon in the mammalian brain. Below is the summary of human results added to the revised manuscript.

“To assess whether temporal transitions between RSFC patterns are also nonrandom in humans, we applied the same analysis to the HCP rsfMRI data from 812 human subjects. […] All these results were highly consistent with our findings in awake rats, suggesting that nonrandom transitions between RSFC patterns are conserved across species and might represent a characteristic feature of the mammalian brain.”

We also discussed the possible reason for higher group-level reproducibility in human data relative to rat data.

“Interestingly, the split-group reproducibility rates were somewhat higher in humans than those observed in rats (both with and without regression of RSFC pattern similarities). […] This notion can be further supported by comparable reproducibility rates if we randomly picked 20 human subjects for each subgroup (reproducibility = 0.91 for human data v.s. reproducibility = 0.86 for rat data), as well as similar reproducibility at the individual level (human data: 0.6( ± 0.05); rat data: 0.57( ± 0.14)).”

Taken together, we believe these results suggest that nonrandom transitions between characteristic RSFC patterns are not only a feature in rodents, but conserved cross species in both humans and rats. Such findings highlight the translational utility of the analysis applied in the present study, which might shed light onto comparative neuroanatomy. Our results have also provided new insight into understanding the spatiotemporal dynamics of spontaneous activity in the mammalian brain.

2) To what extent can the authors exclude movement artifacts. Given the important implications of these temporal shifts, it is important to exclude movements as a cause. Also can the authors identify any other reproducible reasons for the temporal variability?

We appreciate this comment, and completely agree that motion is a critical confounding factor that needs to be teased out. In the revised manuscript, we added more results to rule out the possibility that reproducible temporal transitions between RSFC patterns were attributed to motion artifacts. Specifically, we conducted two additional analyses. First, we re-evaluated the reproducibility of RSFC transitions between two subgroups of rats with relatively high and low motion, respectively. Rats in the first subgroup all had the motion level below the median, assessed by frame-wise displacement (FD). Rats in the second subgroup all had the motion level above the median. Transition matrices were obtained in these two subgroups, respectively. Comparing these two transition matrices yielded the reproducibility of 0.786 (with the regression of seed map similarities, Figure 3—figure supplement 1), which is similar to the reproducibility assessed based on random grouping (0.77, Figure 3B). In addition, the transition matrices from both subgroups were highly consistent with those in randomly divided subgroups (Figure 3B). These results indicate that the high reproducibility of RSFC pattern transitions was not due to head motion.

In the second analysis, we directly compared the motion level between rsfMRI frames involved in RSFC pattern transitions versus those that were not in transitions. All rsfMRI frames we analyzed were categorized into two groups. The first group included frames whose preceding and/or successive frames corresponded to a different RSFC pattern(s). The second group included frames whose preceding and successive frames were the same RSFC pattern. These two groups of rsfMRI frames showed consistent motion levels, quantified by FD values (p = 0.44, two-sample t-test), again indicating that RSFC transitions were not triggered by head motion.

These new results have been added to the revised manuscript.

Reviewer #2:The manuscript by Ma et al. detailed the investigation of the transitions of rsfMRI functional connectivity (RSFC) network patterns in an awake rat. The authors' method of analysis that combines several recent approaches from prior publications – parcellation of RSFC patterns in an awake animal and utilizing graph theory approaches to identify) – is innovative. At present, the findings are novel and such analysis method could potentially be utilized to reveal similar dynamics in human rsfMRI studies. However, the manuscript can be significantly strengthened by further analyses of the current data and the discussions/interpretations improved by better integrating several pieces of prior literature that are presently omitted. I believe that English editing would also further improve this manuscript as there are grammatical errors and at places, the message is not completely clear.1) The authors stated that each rsfMRI run, which comprises of 600 volumes, was divided into rsfMRI frames that were then statistically matched to the 40 characteristic RSFC patterns. The authors then made a comment that 89.9% of the total rsfMRI volumes met the minimum threshold of the spatial correlation coefficient > 0.1. I have two questions regarding these statements:a) How long was each rsfMRI frame? I was unable to find the details in the Materials and methods section.

We apologize for the confusion. Each (3D) rsfMRI frame was 1 sec (time of repetition = 1 sec). We have clarified this issue in the revised manuscript.

b) How well does each rsfMRI frame correspond to one of the 40 characteristic RSFC patterns? It would be best if the authors show and compare the representative 40 RSFC patterns generated from rsfMRI frames and those that were generated from a rsfMRI run (as shown in Figure 2 and Figure 2—figure supplements 1-39). This is because the number of time points included within each rsfMRI run will affect the sensitivity in detecting RSFC patterns and likely influence the subsequent analysis.

We appreciate this comment. The number of time points was the same for each rsfMRI run (540 time points) to ensure the same degree of freedom. In the revised manuscript, we have included the averaged BOLD co-activation patterns from rsfMRI frames that were matched to the reference RSFC patterns (Figure 2 and Figure 2—figure supplement 1), which demonstrated high reminiscence between the BOLD co-activation patterns of rsfMRI frames and the reference RSFC patterns they corresponded to.

2) It is difficult to understand and interpret Figure 5, I believe its quality can be significantly improved. The authors should improve the visualization of the transition network as depicted in Figure 5A and how that translates to Figure 5B. As eLife is a general life and biomedical sciences journal, it would be difficult for readers to grasp the concept of such a complex graph theoretical network analysis just by reading the details in the Materials and methods section. The figure here should be used to best portray a simplistic yet informative summary of the analysis technique and the scientific findings.

Again we apologize for the confusing presentation of Figure 5. We have significantly revised the figure to make it easier to understand. Briefly, Figure 5 showed the community structure of the RSFC pattern transition network. In this network, each node represented a characteristic RSFC pattern, and edges between nodes represented significant transitions between these patterns (permutation test, p<0.05, FDR corrected). Our data showed that this network exhibited prominent community structures with nine modules identified, indicating that RSFC patterns belonging to the same modules had a higher probability to transit between each other than RSFC patterns across modules. The corresponding seed regions of RSFC patterns were color coded based on the community affiliations (Figure 5 inlet).

3) It is still unclear what is the biological significance in the detection of numerous types of transition RSFC patterns. The authors only describe the observation of such transitions but based on the present discussions, it is unclear how valuable or valid these transition patterns are. Moreover, the current discussions made by the authors are insufficient, superficial and sometimes made without citations. For example, the authors state that "bidirectional transitions between striatal and somatosensory/prefrontal RSFC might indicate the presence of bottom-up and top-down processing…" without citing a source. At present, there are numerous studies that interrogated the large-scale effects of optogenetically stimulating striatal, hippocampal and thalamic neurons (Chan et al., 2017; Lee et al., 2016; Leong et al., 2016; Liu et al., 2015; Schmitt et al., 2017). In fact, the authors also characterized the effects of medial prefrontal cortex stimulation (Liang et al., 2015). As such, the authors should make an effort to integrate these pieces of literature or any other relevant works. Note that the literatures suggested here are not exhaustive and that the authors should expand their scope of discussion to better encompass and integrate literature in the interpretation of their findings.

We apologize for omitting important references and lack of in-depth discussion. We have significantly expanded our Discussion to integrate the literature in the interpretation of our results. We appreciate the comment from the reviewer and believe the more in-depth Discussion has significantly improved the quality of our manuscript. Some examples of added discussions are listed below:

“A recent study has shown that low-frequency hippocampal–cortical activity drives brain-wide rsfMRI connectivity, highlighting the pivotal role of hippocampus in RSFC (Chan RW et al. 2017).”

“A recent study investigating the relationship between single neuron spiking activity and brain-wide cortical calcium dynamics found that thalamic spikes could both predict and report (i.e. firing before and after) various types of large-scale cortical activity patterns, which are supported by slow calcium activities (<1 Hz) (Xiao Det al. 2017). These data indicate a pivotal role of the thalamus in transitions between distinct spontaneous brain activity patterns. This result can be further supported by the finding that the ventral thalamus can recruit long-range cortical and subcortical networks and initiate their interactions through low-frequency (1 Hz) activity (Leong AT et al. 2016), and the thalamus can facilitate diverse and brain-wide functional neural integration in a specific spatiotemporal manner (Liu J et al. 2015; Leong ATet al. 2016).”

4) Recently, a paper by Xiao et. al investigated the relationship between single neuron spiking activity and brain-wide cortical calcium dynamics (Xiao et al., 2017). They found that thalamic spikes could produce various types of large-scale cortical activity patterns, which are supported by large-scale slow calcium activities (<1 Hz). More importantly, they demonstrated slow calcium cortical activation patterns in the visual cortical and cingulate regions generated from hippocampal spikes. This finding seems relevant to the observations in the present study as the authors also found the transition between hippocampal networks and visual/cingulate networks. Moreover, Logothetis et al. also showed diffuse responses to hippocampal sharp wave ripple events in multiple cortical areas (e.g., visual and cingulate cortices) (Logothetis et al., 2012). With that in mind, how does the present data fit into the current literature landscape?

Again, we appreciate this insightful and constructive comment. All these studies provide support suggesting the pivotal roles of hippocampus and thalamus in transitions of spontaneous brain activity. We have added more in-depth discussion to the revised manuscript. In addition to the text quoted in response to comment 3. See below for more discussion.

“A recent study has shown that low-frequency hippocampal–cortical activity drives brain-wide rsfMRI connectivity, highlighting the pivotal role of the hippocampus in RSFC transitions (Chan RW et al. 2017). […] Although sharp wave ripples in the hippocampus and RSFC pattern transitions might be from different signal sources, the centralized role of the hippocampus is shared by these two forms of hippocampal-cortical information flow.”

5) In the Discussion subsection “Nonrandom temporal transitions between RSFC patterns”, the authors state that the results of the present study were consistent with previous studies showing that BOLD signal periodically propagated from lateral to medial cerebral cortex (Majeed et al., 2011; Majeed et al., 2009). This statement seems out of place and erroneous as the findings of the present study did not demonstrate any propagation characteristics. Furthermore, the preceding statements in the same Discussion section addressed the reproducibility of the transitions between RSFC patterns, which is out of the context of the works by Majeed et al.

We agree with the reviewer and have removed this sentence in the revised manuscript.

Reviewer #3:The current report examines the spatiotemporal dynamics of spontaneous brain activity in the awake rat, examining whether transitions between RSFC states are nonrandom. While the paper is timely and of potential interest, the overall novelty is limited given previous reports by the same authors (Liang et al., 2015 Ma et al., 2016) and the restricted scope of the analyses conducted.1) Imaging awake rats is advantageous (closer to human RSFC, eliminates effects of anesthesia, etc.), but is more sensitive to head motion. Laumann et al. (2016, Cerebral Cortex) have previously shown that part of the observed dynamics during rest is attributable to head motion. Since motion denoising in this study is lenient (FD exclusion criterion of 0.2 mm as is commonly used in humans) and no global signal regression was applied (which eliminates motion-related artifacts more effectively than ventricle and white matter signal regression alone), it is possible that the non-random transitions between RSFC states are due to head motion. The authors should quantify whether motion events trigger specific transitions. In addition, to more conclusively demonstrate that the results are independent of head motion, the authors can conduct the reproducibility analysis on two sub-groups based on head motion (below an above median FD).

We appreciate this constructive comment and agree motion is a potential confounder and should be teased out with further analysis. In the revised manuscript, we added more results to rule out the possibility that reproducible temporal transitions between RSFC patterns were attributed to motion artifacts. Specifically, we conducted two additional analyses. First, as suggested by the reviewer, we re-evaluated the reproducibility of RSFC transitions between two subgroups of rats with relatively high and low motion, respectively. Rats in the first subgroup all had the motion level below the median, assessed by FD. Rats in the second subgroup all had the motion level above the median. Transition matrices were obtained in these two subgroups, respectively. Comparing these two transition matrices yielded the reproducibility of 0.786 (with the regression of seed map similarities, Figure 3—figure supplement 1), which is similar to the reproducibility assessed based on random grouping (0.77, Figure 3B). In addition, the transition matrices from both subgroups were highly consistent with those in subgroups randomly divided (Figure 3B). These results indicate that the high reproducibility in RSFC pattern transitions was not due to head motion. In the second analysis, we directly compared the motion level between rsfMRI frames involved in RSFC pattern transitions versus those that were not in transitions. All rsfMRI frames we analyzed were categorized into two groups. The first group included frames whose preceding and/or successive frames corresponded to a different RSFC pattern(s). The second group included frames whose preceding and successive frames were the same RSFC pattern. These two groups of rsfMRI frames exhibited consistent motion levels, quantified by FD values (p = 0.44, two-sample t-test), again indicating that RSFC transitions were not triggered by head motion.

These new results have been added to the revised manuscript (Figure 3—figure supplement 1).

2) The finding of non-random transition is novel and will be of interest to the extended community using RSFC to understand functional organization. That said, many aspects of this study are similar to previous studies. Most prominently: RSFC dynamics in awake rats were reported by the authors (Liang et al., 2015). However, the authors do not discuss the relations between the CAP analysis in Liang et al. to the current findings. In addition, the hub analysis replicates some findings from stationary RSFC analysis (Ma et al., 2016), but some findings differ (areas that appear in only one of the analyses). The authors should elaborate on this issue and explain the differences.

We would like to clarify that although RSFC dynamics in awake rodents have been investigated in other and our own studies, they are fundamentally different from the present study.

The present study for the first time characterized the *temporal sequences* between brain connectivity patterns and showed that spontaneous brain activity did not fluctuate randomly, but followed specific orders, while previous dynamic rsfMRI studies, including ours, mainly focused on the spatial characteristics of RSFC dynamics. We believe our results provide new insight into comprehensively characterizing spatiotemporal dynamics of spontaneous brain activity. We have clarified this issue in the revised manuscript.

3) A significant shortcoming of this work is relating it to human RSFC. The manuscript would have been much stronger were the authors able to demonstrate that the observed transitions are present in humans and compare the similarities and differences in the dynamics of RSFC.

Please also see the response to reviewer #1. We completely agree with the reviewer that the manuscript will be significantly stronger with the same analysis applied to human data. In the revised manuscript, we have included new results based on human rsfMRI data collected from 812 subjects in the Human Connectome Project (HCP) using the same approach. Briefly, we found that, like our results in rats, transitions between brain connectivity patterns were highly reproducible in humans, suggesting that transitions between characteristic resting-state functional connectivity (RSFC) patterns are nonrandom. These data demonstrated that this feature was well conserved across species in rats and humans, and might represent a general phenomenon in the mammalian brain. Below is a brief summary of human results but please see more details in the revised manuscript.

The reproducibility between two subgroups (random split) was 0.9955 (without the regression of seed map similarities), and 0.9954 (with the regression of seed map similarities), respectively. To assess the reproducibility at the individual level, the correlation between the transition matrix of each human subject versus the group-level transition matrix was calculated. The mean correlation ( ± SD) was 0.60 ( ± 0.05).

Taken together, we believe these results suggest that nonrandom transitions between characteristic RSFC patterns are not merely a feature in rodents, but conserved cross species in both humans and rats. Such findings highlight the translational utility of the analysis applied in the present study, which might shed light onto comparative neuroanatomy. Our results have also provided new insight into understanding the spatiotemporal dynamics of spontaneous activity in the mammalian brain.

4) In the Discussion, the authors conclude that RSFC patterns obtained based on these parcels are characteristic and provide a comprehensive representation [to] all ~6000 RSFC patterns. This conclusion goes beyond the data. The number of 40 parcels was chosen by the authors in their previous report (Ma et al., 2016) as an example for a low-dimensionality parcellation, it's possible the high dimensionality parcellation will better represent the ~6000 voxels of the rat brain, especially when taking into account the fact that the rat brain is known to have more than 40 regions. To support their conclusion, the authors need to demonstrate that the high-dimensionality parcellation from Ma et al., 2016 (130 parcels) does not overperform the 40 parcels.

We agree with the reviewer and acknowledge that the number of 40 was arbitrarily selected as an example of low-dimensionality parcellation. Similar analysis can be applied using other parcel numbers. In the revised manuscript, we removed the word ‘comprehensive’. We have also pointed out this issue in the section of Potential Limitation. As we stated, corresponding a rsfMRI frame to its most similar reference RSFC pattern is only an approximation for the purpose of investigating spatiotemporal dynamics of spontaneous brain activity. To mitigate this issue, we ensured that the similarity between each rsfMRI frame and its best matched RSFC pattern was statistically significant (p< 10^-13^). In addition, in the revised manuscript we added averaged rsfMRI frames matched to each characteristic RSFC pattern, which demonstrated high reminiscence between the BOLD co-activation patterns of rsfMRI frames and the RSFC patterns they corresponded to. As a result, we believe the characteristic RSFC patterns selected can well represent rsfMRI frames.

5) The authors fail to cite many recent publications relevant to the current work, including awake rodent fMRI from other groups (e.g., Bergmann et al., 2016; Chang et al., 2016; Yoshida et al., 2016), dynamic RSFC in mice (Grandjean et al., 2017) and spatiotemporal dynamics of intrinsic signals (Mohajerani et al., 2013).

We apologize for missing these important references. In the revised manuscript, we have included these papers.

[Editors' note: the author responses to the re-review follow.]

Essential revisions:1) The addition of human data is important and increases the impact of this study. However, the high correlation values between the 2 subgroups (>0.995) and the fact that significant transitions are very sparse, bring to question the role of statistical thresholding. Since only a small fraction of the transitions are significant, thresholding in-and-of-itself can artificially increase correlation values between subgroups. In addition, thresholding can explain the higher reproducibility in humans relative to rats (human matrices seem sparser). The authors should examine the matrices with all of the raw data (i.e., without applying a threshold).

We apologize for the confusion. The matrices of human data we showed in the previous submission were unthresholded. They were indeed very sparse, attributed to a much higher number of RSFC patterns involved relative to rat data (humans: 333 RSFC patterns; rats: 40 RSFC patterns), which led to low- (or zero-) probability transitions between quite a lot of RSFC patterns. We did not include any thresholded results in our previous submission due to high computational intensity of permutation tests. However, we included new results from thresholded data (see more details in the response to comment 2).

2) Along the same lines, in order to draw strong similarities/parallels between the human and rat rsfMRI data, the authors included the analysis outcome of the reproducibility matrix of RSFC patterns in humans (Figure 8) to demonstrate that the transition of RSFC patterns are nonrandom and echo the observations found in the awake rat brain (Figure 3). However, the authors should also include additional analysis outcomes that were generated from the human rsfMRI dataset as outlined by the analysis flowchart in Figure 1, such as RSFC pattern transition matrix (Figure 4) and graph theoretical analyses (Figures 5 and 6). It would appear from the flowchart that most of these analyses were already performed in order to generate the reproducibility matrix (Figure 8). Only then, a stronger case could be made by comparing multi-dimensional data to draw similarities between the rat and human brain, particularly in RSFC pattern transitions across time. On a minor note, it may also be of interest to look at the parcellated regions in the human brain using the method developed by the authors as there are numerous recent papers that attempt to improve parcellation procedures (Glasser et al., 2016).

In the revised manuscript, we added new results of human data including RSFC pattern transition matrix (Figure 9) and graph theoretical analysis (Figure 10). The RSFC pattern transition matrix was obtained by applying the same permutation test that identified transitions statistically above chance. We found that, like rats, RSFC pattern transitions in humans more frequently occurred within the same brain system, but considerable cross-system transitions were also observed. We further calculated the hubness of individual nodes in the human RSFC transition network, and found multiple hubs in separate brain systems including default-mode, cingulo-opercular, dorsal attention, ventral attention, fronto-parietal, parietal memory and visual networks. Intriguingly, virtually all hubs were integrative networks (with the only exception of the visual network) that are known to integrate information from multiple other areas (e.g. sensori-motor systems). Our data suggest that these integrative networks played a pivotal role in human RSFC pattern transitions. These findings highlight the translational utility of the analysis applied in the present study, which might shed light onto comparative neuroanatomy. Our results have also provided new insight into understanding the spatiotemporal dynamics of spontaneous activity in the mammalian brain.

Unfortunately, our computers do not have enough memory to conduct the parcellation of the human brain using the same method we developed in the rat due to a much larger matrix size required for the human data.

3) The possibility that motion-related artifacts may explain some of the data continues to be a major potential concern, and the authors need to address this concern in a systematic manner.3a) Based on the relatively modest difference between random and motion-based splitting (0.77 vs. 0.786) the authors claim that the motion-based reproducibility analysis suggested indicates that the high reproducibility in RSFC pattern transitions was not due to head motion. However, this inference is not supported by a statistical test. A permutation test in which the authors calculate reproducibility for many random divisions to subgroups and compare the distribution of correlations would allow this type of inference. In addition, the authors described the results of this analysis only after the regression of seed map similarities, but the supplementary figures should replicate the full Figure 3 and show the results both with and without this regression. Moreover, the authors should describe the FD values (mean+SD) of the subgroups above and below the median FD so the reader will be able to evaluate the data quality.

We appreciate this constructive suggestion. In our revised manuscript, we added new results of a permutation test to examine whether the reproducibility using motion-based group division was statistically different from that based on random divisions. Specifically, we repeated random divisions to subgroups 10000 times. Mean reproducibility ( ± SD) across all 10000 trials was 0.876 ( ± 0.010) and 0.791 ( ± 0.017) without and with regression of seed map similarities, respectively. Figure 3—figure supplement 2 showed the distribution of the reproducibility between randomly divided subgroups across all trials. This data demonstrated that the reproducibility using motion-based division to subgroups was not statistically different from the reproducibility based on random division regardless whether RSFC pattern similarities were regressed out (p = 0.75) or not (p = 0.67). In our revised manuscript (Figure 3—figure supplement 1), we reported the reproducibility results both with and without regression of RSFC pattern similarities.

We also reported the FD values of the subgroups above and below the median FD as suggested. The mean ( ± SD) FD of the below-median subgroup was 0.037 ( ± 0.024) mm, and the mean ( ± SD) FD of the above-median subgroup was 0.054 ( ± 0.033) mm.

3b) In their second analysis, the authors examine whether transitions predict head motion. However, this approach is not very sensitive since it tests whether all transitions are caused by motion and not whether motion causes some of the transitions. The authors should test whether transitions are more likely to follow head motion and if such transitions are random across areas or affect specific areas preferentially.

We agree that the analysis mainly tested whether all transitions are caused by motion. To further address this issue, in the revised manuscript we measured the head motion (i.e. FD) during each RSFC pattern transition and compared the mean head motion level for each transition sequence and the occurrence count of this transition sequence. Specifically, we calculated the mean FD for transitions between every two RSFC patterns. This calculation yielded a 40 × 40 matrix, in which each element quantified the mean FD for each transition sequence (e.g. element (i,j) of this matrix measured the mean FD for the transition from RSFC pattern i to RSFC pattern j). Our data showed that the correlation between this transition FD matrix and the RSFC pattern transition matrix was minimal (r = -0.034), suggesting that the mean head motion level during each transition sequence did not predict the occurrence count of this transition sequence. This result further supports that RSFC pattern transitions were independent of head motion, and RSFC transitions did not follow head motion.

3c) An alternative approach to address this issue is lowering the motion censoring threshold. The current one (0.2 mm) is taken from the human fMRI literature, and might not be suitable for the rat brain. Similar results with FD thresholds of 0.15 mm, 0.1 mm and 0.05mm can indicate that the transitions are motion independent.

We agree that showing robust results for a more stringent censoring threshold will further strengthen the study. In the revised manuscript, we reanalyzed our data using a more stringent censoring threshold (FD < 0.1mm). At this threshold, very similar RSFC pattern transition matrices were obtained. The correlations between the RSFC pattern transition matrices at FD < 0.1 mm and those at FD < 0.2 mm were 0.83 and 0.88 with and without regression of RSFC pattern similarities, respectively, suggesting that our results were robust and insensitive to the motion censoring threshold applied (Figure 3—figure supplement 3). We did not use an even more stringent threshold at FD < 0.05 mm because significant portion of data were discarded at this threshold.

4) Along the same lines, no description in the motion scrubbing methods is given for the human data. Did the authors exclude any motion-related artifacts?

The ICA+FIX pipeline (Griffanti L et al. 2014; Salimi-Khorshidi G et al. 2014) was used for cleaning head motion, and this process was already performed in the HCP pipeline in ‘HCP extensively preprocessed 3T rsfMRI data’. We have clarified this issue in the Materials and methods section in the revised manuscript.

5) The amount of raw data presented is considerable, but the number of second-level quantitative analyses is limited. This shortcoming makes the manuscript difficult to read and not very attractive for a broad audience. Below are some suggestions to address this concern:5a) In Figure 2, the addition of the distribution of correlation between Seed maps and Average of rsfMRI frames for the 40 seeds will be highly advantageous. Currently, the reader has to visually evaluate 40 seeds in Figure 2-figure supplement 1. Note that parts of Figure 2—figure supplement 1 are cropped (left and lower aspects).

In the revised manuscript, we have added the quantification of similarity between seed maps and average of rsfMRI frames, measured by their spatial correlations, for all 40 seeds. We have also fixed the cropped parts of Figure 2—figure supplement 1.

5b) In Figure 4, the authors present a 40x40 matrix. It is impossible to properly evaluate it. Others have used grouping to categories (based on the axes / brain systems) to assist the reader in evaluating which brain systems are characterized by transitions.

We have added axes in Figure 4 to indicate the brain system that each row and column belong to.

5c) Figure 6 depict seed maps that were already presented in previous figures. This redundancy is unwarranted. The authors can plot the hub score of the different 40 seeds and annotate brain systems.

We have revised the figure as suggested.

5d) Figure 7 depicts, again, seed maps that were previously shown. The authors need to find a more compact presentation.

We have changed the presentation of Figure 7 by replacing the seed maps with nodes that include the pattern number shown in Figure 2 and the brain system of the seed (Please see Figure 7 in the revised manuscript).

5e) Figure 8 depicts the human matrix. If the rat 40x40 matrix was challenging to evaluate, the human 333x333 matrix is virtually unreadable. The authors need to find a visual representation proper for this data via alternative visualization methods (perhaps a scatterplot will be a better way to compare between the 2 subgroups).

We have added scatterplot panels to the figure as suggested. We have also added more annotations to better present the figure.

6) The authors need to better explain the significance/innovation of the present study, given that previous publications from the authors' laboratory and others have already demonstrated characteristics of RSFC pattern transitions across time (Chang et al., 2016; Liang et al., 2015; Liu and Duyn, 2013; Liu et al., 2015). It is clear that these already published studies did not extensively or exclusively document the transition of multiple RSFC patterns across time, but the authors need to better explain why the present study is not just an incremental advance.

We have better clarified this issue in the revised manuscript. All these previous studies in this line, including ours, focused on revealing the spatial organization of RSFC patterns that were non-stationary, but none of these studies examined the temporal relationship between these RSFC patterns. In other words, previous studies only established the existence of RSFC temporal transitions, but it was unclear whether these transitions were random or organized in an orderly manner. The innovation of the present study is to elucidate the temporal organization of RSFC pattern transitions in both humans and rats. We found that temporal transitions between spontaneous brain activity were not random, but follow specific orders, and this result was consistent in both awake rats and humans.